# Decreasing Trend of Geohazards Induced by the 2008 Wenchuan Earthquake Inferred from Time Series NDVI Data

**Zhongyun Ni** [1,2,3], **Zhenyu Yang** [1,*], **Weile Li** [3], **Yinbing Zhao** [2,3] and **Zhengwei He** [3]

1   College of Resource Environment and Tourism, Capital Normal University, Beijing 100048, China; nizhongyun2012@mail.cdut.edu.cn

2   College of Tourism and Urban-Rural Planning, Chengdu University of Technology, Chengdu 610059, China; zhaoyinbing06@cdut.cn

3   State Key Laboratory of Geohazard Prevention and Geoenvironment Protection, Chengdu University of Technology, Chengdu 610059, China; liweile08@mail.cdut.edu.cn (W.L.); hzw@cdut.edu.cn (Z.H.)

*   Correspondence: zhenyu.yang@cnu.edu.cn; Tel.: +86-10-6890-2971

**Abstract:** The occurrence of aftershocks and geohazards (landslides, collapses, and debris flows) decreases with time following a major earthquake. The 12 May 2008 Wenchuan Earthquake in Sichuan, China, provides the opportunity to characterize the subsequent spatiotemporal evolution of geohazards. Following the 12 May 2008 Wenchuan Earthquake, the incidence of geohazards first increased sharply, representing a "post-earthquake effect", before starting to decrease. We compared the spatial distribution of the area affected by vegetation damage (AVD) triggered by large and medium-scale geohazards (LMG). We studied the interval prior to the 12 May 2008 Wenchuan Earthquake (2001–2007), the co-seismic period (2008), and the post-earthquake interval (2009–2016) and characterized the trend of decreasing geohazards at a macro scale. In vegetated areas, geohazards often seriously damage the vegetation, resulting in pronounced contrasts with the surrounding surface in terms of color tone, texture, morphology, and Normalized Difference Vegetation Index (NDVI) which are evident in remote sensing images (RSI). In principle, it is possible to use the strong positive correlation between AVD and geohazards to determine indirectly the resulting vegetation and to monitor its spatiotemporal evolution. In this study we attempted to characterize the process of geohazard evolution in the region affected by the 12 May 2008 Wenchuan Earthquake during 2001–2016. Our approach was to analyze the characteristics of areas with reduced vegetation coverage caused by LMG. Our principal findings are as follows: (i) Before the Wenchuan Earthquake (during 2001–2007), there was no evidence for a linear increase in the number of LMG with time; thus, the geological environment was relatively stable and the geohazards were mainly induced by rainfall events. (ii) The 12 May 2008 Wenchuan Earthquake was the main cause of a surge in geohazards in 2008, with the characteristics of seismogenic faults and strong aftershocks determining the spatial distribution of geohazards. (iii) Following the 12 May 2008 Wenchuan Earthquake (during 2009–2016) the incidence of geohazards exhibited an oscillating pattern of attenuation, with a decreasing trend of higher-grade seismic intensity. The intensity of geohazards was related to rainfall and seismogenic faults, and also to the number, magnitude and depth of new earthquakes following the 12 May 2008 Wenchuan Earthquake. Our results provide a new perspective on the temporal pattern of attenuation of seismic geohazards, with implications for disaster prevention and mitigation and ecological restoration in the areas affected by the 12 May 2008 Wenchuan Earthquake.

**Keywords:** time series NDVI; spatiotemporal evolution; area affected by vegetation damage (AVD); 12 May 2008 Wenchuan Earthquake

## 1. Introduction

Giant earthquakes usually cause geohazards, fires (e.g., the San Francisco Earthquake, Mw 8.3, 18 April 1906), tsunamis (e.g., the Indian Ocean Tsunami, Mw 8.5, 26 December 2004), and other secondary hazards [1–4]. The main shock results in a rapid release of crustal stress, which is often accompanied by aftershocks, and the frequency of aftershocks then decreases with time, which is often termed Omori's Law of Earthquakes [5]. In the years following a major earthquake, secondary geohazards in the crustal rupture zones or disaster areas may consist of high-intensity activity for a certain length of time, under the stress of heavy rainfall events and other factors. In some cases, secondary geohazards may persist for a relatively long time interval: for example, they persisted for 40 years in the case of the Kanto Earthquake (1 September 1923, Mw 8.1) [6–10]; in the case of the Chi-Chi Earthquake (21 September 1999, Mw 7.6), high-intensity landslides occurred for at least 6 years [11–13]; but in the case of the Kashmir Earthquake (8 October 2005, Mw 7.6), secondary geohazards persisted for no more than 2 years [14,15]. Under heavy rainfall, the number and scale of certain geohazards is large [6,11,13,16]; however, this may not be case where the influence of precipitation is less [14,15]. With the progressive reduction of the amount of unconsolidated materials, the occurrence of new geohazards (e.g., landslides, slope instability and debris flows) decreases gradually; in addition, geohazard activity in earthquake-stricken areas (ESA) also decreases gradually. Remote Sensing images (RSI) and field investigations have been used to compare changes in the number, volume and area of surface rupture caused by geohazards before and after an earthquake, or to compare the output of river sediments before and after an earthquake [8,9,11,13–16]. Such studies are designed to characterize the stability of the environment disturbed by geohazards, given the absence of any further large-scale geohazards after heavy rainfall.

Following the Wenchuan Earthquake (12 May 2008, Mw 7.9, USGS), several studies were conducted which revealed a decreasing temporal trend of the resulting geohazards [17–23], but the spatiotemporal characterization of the process was limited. Several authors have predicted that following a major earthquake, landslides and rockfalls would persist for 5–10 years and debris flows for 10–20 years, and that the degree of vegetation coverage of the slopes is the main factor determining slope stabilization [17–19]. Landslides can be easily triggered by storms, resulting in the blocking of rivers and forming a barrier lake. Models have been used to estimate the area and volume of landslides under rainfall events of different intensities, and it has been predicted that landslides and debris flows would be maintained for at least 10 years [20,21]. Based on the distribution characteristics of high-position debris flows (HPDF), it has been predicted that post-earthquake geohazards would persist for 15 years and that it would take several decades for conditions to recover to the pre-earthquake level [22]. It has been pointed out, that the ESA experienced a period of high-intensity geohazards triggered by relatively low threshold rainfall events [23]; however, as the regional geological environment tended to stabilize, the threshold of rainfall events needed to trigger new landslides would increase. It has been suggested that geohazards would persist for 20–25 years after the Wenchuan Earthquake, decreasing in the form of a series of oscillations with a recurring peak every 4–5 years, and eventually returning to the pre-earthquake level [24,25]. Several researchers [21,26] have used a model based on the relationship between area and volume of geohazards to estimate the total volume of landslide materials generated in 2008, and contrasted it with the output of debris flows in a small watershed located in the most seriously affected areas; thus, they were able to determine the evolutionary trend and life span of the occurrence of geohazards within the entire ESA. However, the results may have overestimated the duration of the process of evolution of geohazards, because the ultimate amount of landslide erosion is inevitably much less than the total amount of potential landslide material.

In this study, we attempted to analyze the decreasing temporal trend of geohazards from the perspective of the area affected by vegetation damage (AVD), with the hope that the findings could be applied to other areas with strong tectonic activity. Specifically, we sought to answer the following questions: What was the relationship between AVD and the area of geohazards? Could MODIS-NDVI products with the highest spatial resolution (250 m) be used to monitor geohazards? What were

the characteristics of the trend of decreasing geohazards in specific localized contexts, such as at fault intersections?

Earthquakes are a type of earth surface deformation [27]. Earthquakes that induce surface ruptures in different geomorphic units inevitably cause changes in vegetation coverage [28]. The 12 May 2008 Wenchuan Earthquake triggered a huge amount of rockfalls, landslides and debris flows which formed barrier lakes [18,29–32]. These events caused a reduction in vegetation cover, disturbance of the litter layer, destabilization of tree roots, degradation of vegetation-soil systems, decline in ecosystem functioning, landscape fragmentation, and decreases in the density of the forest canopy [33–36]. AVD typically produces a significant contrast to the surrounding background, in terms of color tone, texture, morphology, and the spectral and NDVI characteristics of RSI [37,38]. Interpretation of the results of medium-high spatial resolution RSI can demonstrate a high positive correlation between AVD and the areas affected by co-seismic geohazards [33,39]. Thus, the availability of AVD data with a large temporal range can potentially be used to reflect the evolution of geohazards. If so, it can provide a new methodology for monitoring geohazards from the perspective of damaged vegetation.

Compared with field survey and fixed-point monitoring, remote sensing technology has the advantages of a wide monitoring range, flexibility of data acquisition time, and the comprehensive utilization of multiple data sources. Analysis of the spatial distribution characteristics of AVD caused by co-seismic geohazards has been conducted via the visual or computer interpretation of medium-high spatial resolution RSI (e.g., Landsat, 30 m; CBERS, 19.5 m; ASTER, 15 m; ALOS, 10 m; IRS-P5, 5.8 m; SPOT, 2.5 m; Cartosat-1, 2.5 m; FORMOSAT-2, 2 m; IKONOS, 1 m; Quick-Bird, 0.61 m; World-View I/II, 0.5 m), or by the use of Unmanned Aerial Vehicle (UAV) images (0.3 m), or by field investigation. This has been combined with the analysis of the relationship between the AVD and the geological environment using auxiliary data [33–36,40–42].

The detailed characterization of the process of geohazards evolution can only be realized by the availability of continuous monitoring data spanning at least 5–10 years. Thus, the limitations of medium-high spatial resolution optical RSI [43], such as low availability caused by clouds and rainfall, low frequency data output due to the long revisiting cycle, relatively narrow photographic scope and the time- and labor-intensive nature of data processing, make it difficult to use the approach in practice. Among the existing RSI, MODIS has the advantage of temporal resolution because MODIS Terra and MODIS Aqua satellites can observe the same land surface twice per day. The co-seismic and post-earthquake geohazards in our study area occurred in the form of 'chains' and were of large-scale and numerous [18,21,22,25,32,44,45]. The distribution of AVD caused by co-seismic geohazards has been monitored using MODIS-NDVI with a 250 m spatial resolution, together with analysis of the restoration of vegetation during the years following the 12 May 2008 Wenchuan Earthquake [38,46–53]. However, due to specific research aims, the distribution of co-seismic geohazards was only monitored in 2008 [38,46,54], and no attempt was made to conduct monitoring of multi-year time series.

The present research project was initiated by the wish to characterize the decreasing trend of large and medium-scale geohazards (LMG) over time in an extremely severe ESA, with attention being paid to specific locations (e.g., fault zones with strong tectonic activity). It was hoped that the results would provide a basis for policy formulation for disaster prevention, mitigation, and restoration via ecological engineering.

## 2. Materials and Methods

### 2.1. Study Area

#### 2.1.1. Geographical Background

The study area comprises 10 Counties/Cities (the Counties of Wenchuan, Beichuan, Qingchuan, Maoxian, Pingwu and Anxian; and the Cities of Mianzhu, Shifang, Dujiangyan and Pengzhou) with an area ~30,000 km$^2$ and is located in the northern part of Sichuan Province. The terrain is step-like from east to west. From the Chengdu Plain to the Western Sichuan Plateau, the altitude ranges from

400–6000 m, and the terrain comprises deep valleys and steep mountains. It is within the main area of influence of the Longmenshan Fault Zone; the rear fault extends from Wenchuan to Maoxian, the central fault from Yingxiu to Beichuan, and the front fault from Pengzhou to Guanxian [19]. As a result of the continuous extrusion of the Qinghai-Tibet Plateau, tectonic activity in the region was frequent during the Quaternary [55]. Before the 12 May 2008 Wenchuan Earthquake, eight earthquakes with a magnitude > 7.0 on the Richter scale occurred in areas adjacent to the Longmenshan Fault Zone [56]. The Diexi Earthquake (7.5 on the Richter scale) occurred near Maoxian County in 1933 and was the most serious and best-documented recent earthquake, destroying an ancient town and triggering floods which killed more than 2000 people [57].

After the 12 May 2008 Wenchuan Earthquake, 10 Counties/Cities were classified as the most severely damaged areas with the highest numbers of casualties and property loss and the most severe ecological destruction [58]. Based on the severity of earth surface ruptures and the destruction of buildings caused by the 12 May 2008 Wenchuan Earthquake, the seismic intensity level was divided into 6 classes (VI-XI) [59]. More severe damage was caused by higher levels of seismic intensity. The area that experienced an intensity of XI represented more than 40% of the entire affected area [52]. Five localities (Gengda-Longchi, Hongkou-Yinchanggou, Hongbai-Chaping, Leigu-Chenjiaba, and Nanba-Donghekou) would continue to experience a high incidence of geohazards after the Wenchuan Earthquake [24,25]. In the present study, we selected these 10 Counties/Cities as the research area and 5 localities as the key studied area (Figure 1), and analyzed the spatiotemporal evolution of geohazards during 2001–2016 and attempted to characterize the decreasing temporal trend of geohazards evolution following the 12 May 2008 Wenchuan Earthquake.

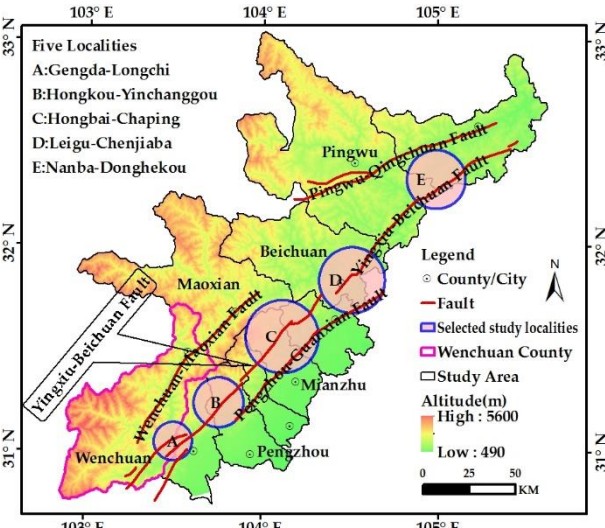

**Figure 1.** DEM (Digital Elevation Model) of the study area in Sichuan Province, China. The black polygon represents the study area; the area delineated by the magenta line represents Wenchuan County; the pink circles labelled A–E are the five studied localities; the red lines are faults. The small dotted circles are County/City locations. The five key studied localities are as follows, A: Gengda (Wenchuan County)-Longchi (Dujiangyan City) locality; B: Hongkou (Dujiangyan City)-Yinchanggou (Pengzhou City) locality; C: Hongbai (Shifang City)-Chaping (Anxian County) locality; D: Leigu (Beichuan County)-Chenjiaba (Beichuan County) locality; E: Nanba (Pingwu County)-Donghekou (Qingchuan County) locality.

### 2.1.2. Characteristics of Geohazards

The occurrence of geohazards is closely related to lithology, geological structures, seismic activity, hydrometeorological factors, and topographical features (e.g., the distribution of rivers, valley density, topographic relief, and the length and steepness of slopes). It is also related to human factors such as

engineering activity associated with mining and road excavation along valley slopes [60,61]. Active structures increase the vulnerability of the geological environment [62,63].

Before the 12 May 2008 Wenchuan Earthquake, the outbreaks of almost all the geohazards in the study area were related to heavy rainfall events and long-term continuous precipitation [64]. Rainfall is abundant in the region and is concentrated during June–October. The annual number of rainstorm days in Beichuan and elsewhere exceeds 10 [65]. Influenced by rainfall, the tributary watersheds were the most frequent destination of small-scale debris flows, while the main ditch watersheds experienced large- to medium-scale debris flows with the low frequency of one outbreak every 30–50 years [66]. In Sichuan Province, with a total area of ~164,000 km$^2$, the Western Sichuan Plateau and the Southwest Sichuan Mountainous Area are at high risk of geohazards [67]. Among the 5770 geohazards in Sichuan Province during 1999–2000, the numbers of large- and medium-scale landslides, rockfalls and debris flows were 1213, 550, and 558, respectively, representing 40.75% of the total number of geohazards [60]. Rockfalls, landslides and debris flows were the major types of geohazards, and mostly occurred along the fault zones or river valleys [61]. Landslides/rockfalls with output volumes of $>1000 \times 10^3$ m$^3$, $100–1000 \times 10^3$ m$^3$, $10–100 \times 10^3$ m$^3$, and $<10 \times 10^3$ m$^3$, are defined as extra large-scale, large-scale, medium-scale and small-scale, respectively. Debris flows with material outputs of $>500 \times 10^3$ m$^3$, $100–500 \times 10^3$ m$^3$, $10–100 \times 10^3$ m$^3$ and $<10 \times 10^3$ m$^3$ are defined as extra large-scale, large-scale, medium-scale and small-scale, respectively.

The co-seismic geohazards induced by the 12 May 2008 Wenchuan Earthquake (e.g., rockfalls, landslides, and destabilized slopes [45]) were located across an area exceeding 100,000 km$^2$. The interpretation of multi-source RSI with 197,481 polygons indicated that the total landslide area was ~1159 km$^2$ [68]; thus, the landslide characteristics were large in terms of scale, quantity, and mass distribution and caused widespread devastation [45]. The area of AVD caused by co-seismic geohazards was ~1249 km$^2$ [39]. The 60,109 landslides were identified using points [69], and 59,108 landslides using polygons [31]. The area devastated by the 12 May 2008 Wenchuan Earthquake was ~1679 km$^2$ [70], and the area devastated by 80% of the large-scale landslides was concentrated on both sides of the rupture zones within the range of 5 km [71]. They were mainly developed in river gullies and valleys below 1500–2000 m with significant "terrain magnifying", "hanging wall" [72] and "segment locking" effects [71]. Thus, there was a linear distribution along the direction of fault ruptures which was consistent with the strike of the fault zones in a southwest-northeast direction. The geohazards occurred on slopes with angles ranging from 20–50°. The types of geohazards (rockfalls or landslides) were determined by lithology and slope structure [30,32,68,72,73].

Following the 12 May 2008 Wenchuan Earthquake, local cumulative precipitation reached 468 mm within three days (9–11 July, 2013) [74], which triggered new geohazards in the form of debris flows and resulting barrier lakes [22,29,75,76]. In Yingxiu-Wenchuan County, debris flows occurred in more than 200 gullies on 10 July, 2013; the largest was the Qipan gully debris flow with a volume of $1.0 \times 10^6$ m$^3$ [77]. These phenomena occurred within a specific sequence or chain: landslips, landslides, debris flows ⇒ landslide lakes ⇒ outburst floods [19]. From May 2008 to April 2015, 660 new landslides occurred near the epicenter, outside the co-seismic landslide areas [23].

### 2.1.3. Characteristics of Vegetation-Damaged Areas

Influenced by the southwest and southeast monsoon, the climate of the study area is warm and humid which promotes abundant vegetation growth [78]. The vegetation types are varied and include coniferous forest, broadleaved forest, coniferous and broadleaved mixed forest, shrubland and grassland [79].

Geohazards such as rockfalls, landslides and debris flows often cause severe damage to the ecological environment. The most direct manifestation of this is the effects on the vegetation of woodland, sparse woodland, shrubland, grassland and farmland, which may be eradicated or buried, or in the case of trees-collapsed. The occurrence of large-scale geohazards may result in entire forested areas being destroyed, forming a "forest window" [35]. Trees may be tilted, broken, dislodged and

killed. Shrubland or grassland may be buried or slipped and piled up in the lower part of slopes. Under the action of rainfall, litter and plant debris, combined with debris flows materials, may move and damage the vegetation and/or farmland downslope and downstream [36]. Even if the vegetation in some areas is not destroyed directly by immediate geohazards, the occurrence of ground fissures can weaken and dislodge rocks and soils which were previously stabilized by plant roots; the subsequent infiltration of precipitation in these areas may cause them to become unstable, causing further damage to the vegetation on the slopes [34].

The area affected by vegetation damage (AVD) refers to the area where the vegetation coverage has been affected by the occurrence of geohazards. From RSI [37], the AVD appeared as "scars" and was distinctly different from the background of areas with relatively dense vegetation coverage. The color tone of the AVD was clearly highlighted and was consistent with that of bare rock or soil. The texture was influenced by lithology and the mode of landslide initiation. In the areas denuded by landslides, the land surface was relatively uniform and fine-textured; while in the areas where large particles or a diverse range of debris types had accumulated, the texture was rough. The various patterns were related to the types of geohazards: fan-shaped, dustpan-shaped, cup-shaped, strip-shaped, circular or irregular. The accumulation of loose materials often formed irregular shapes in the downstream areas of rivers and the lower parts of valleys and roads. The affected areas not only showed the same spectral characteristics as bare rock or soil, but they also showed an abrupt decrease in NDVI values in areas with a previously high degree of vegetation coverage [38].

The following points are noteworthy: (i) The geohazards occurring before the 12 May 2008 Wenchuan Earthquake were caused by geological activity. In this study, these prior geohazards which caused vegetation damage are those which occurred during 2001–2007; they were mainly rockfalls, landslides and debris flows. (ii) We define co-seismic geohazards as the deformation or destruction of geological bodies caused by seismic activity; they are usually rockfalls and landslides. The occurrence of co-seismic geohazards was almost coincident with the occurrence of the 12 May 2008 Wenchuan Earthquake. (iii) Secondary geohazards following the 12 May 2008 Wenchuan Earthquake, which occurred in areas of high seismic intensity (usually in the form of rockfalls, landslides and debris flows occurring in groups or chains) were affected by aftershocks and rainfall events. The related interval of vegetation damage was 2009–2016. Geohazards included co-seismic geohazards and post-earthquake secondary geohazards. Geohazards, such as potentially unstable slopes, ground fissures and ground collapse, are not considered here due to the small extent of the AVD they caused, comparing to the pixel-scale or region-scale of the RSI used in the study. They have an insignificant effect on the 'law of geohazards' that we attempt to determine for the area affected by the 12 May 2008 Wenchuan Earthquake.

*2.2. Materials*

In this study, the 368 MOD13Q1 images (see the Supplementary Materials for the download link) [80] were used as a remote sensing data source to extract the AVD. They were 16-day composite MODIS-NDVI products with 250 m resolution spanning 2001–2016, with Julian days from 001–353. They were freely available from the Land Processsinged Distributed Active Center (LP DAAC). High spatial resolution RSI (e.g., Gao-Fen, Google Earth) and Landsat-5 TM, Landsat-7 ETM+ and Landsat-8 OLI (Operational Land Imager) (see the Supplementary Materials for the download link) and the points of geohazards by field investigation were used to verify the AVD in the five localities and valleys, or in accessible locations.

The earthquake data from the USGS (see the Supplementary Materials for the download link) were used to analyze the relationship between AVD and geohazards. The precipitation data from Dujiangyan Observatory Station (elevation ~698 m) for the interval from January 2001 to December 2016 was used to analyze the relationship between the occurrence of geohazards and the amount of rainfall.

*2.3. Methods*

From the perspective of identifying AVD, the long-term monitoring of the occurrence of LMG in the areas affected by the 12 May 2008 Wenchuan Earthquake, and for characterizing the evolution of geohazards, was based on two hypotheses. First, the study area had a high vegetation coverage before the 12 May 2008 Wenchuan Earthquake, and the AVD could reflect the occurrence of LMG. Second, the AVD had a high positive correlation with the area of geohazards. The specific steps were as follows:

(i)　　Analyze the feasibility of monitoring LMG by time series of MODIS-NDVI.
(ii)　　Select the best method of building and fitting the time series of MODIS-NDVI.
(iii)　Extract the spatial distribution of LMG for 2001–2016.
(iv)　Analyze the spatiotemporal evolution of LMG during 2001–2016 and characterize the decreasing temporal trend of LMG during the post-earthquake period of 2008–2016.

A flow chart illustrating the process of extracting the area of vegetation caused by geohazards is shown in Figure 2.

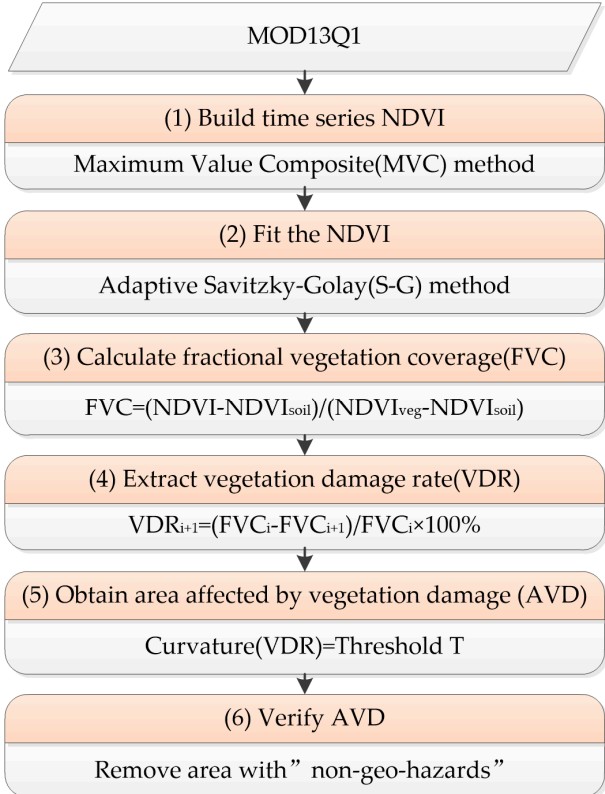

**Figure 2.** Flow chart of the steps involved in extracting the area affected by vegetation damage caused by geohazards.

2.3.1. Building an NDVI Time Series

Maximum Value Composite (MVC) is a widely used synthetic method to eliminate the disturbance caused by clouds, other atmospheric effects, and outliers or missing data, caused by variations in solar altitude angle and sensor observation angle. MOD13Q1 has 23 date scenes per year. Monthly-NDVI products were generated using the MVC method and two NDVI data were combined into a one-month product. As a result, the 12 data could be produced for one year and 192 data could be generated for 16 years. One file could be synthesized from 192 files using the layer stacking function of the ENVI 5.3 software.

### 2.3.2. Fitting the NDVI

NDVI is the most important index for studying vegetation dynamics [81]. The decrease in NDVI values may be caused by vegetation disturbances that are emergency events (such as geohazards, fires and floods), natural vegetation replacement, and human activities. A commonly used filtering technique based on pixel spectral features is the adaptive Savitzky-Golay (S-G) method [82,83]. By fitting the NDVI time series using this technique, background noise caused by weather and other factors can be effectively eliminated, enabling the vegetation growth information to be highlighted. The mean value is used to replace the original value within a sliding window, after that, the quadratic polynomial method is utilized to fit the NDVI time series.

### 2.3.3. Calculation of FVC

Fractional vegetation coverage (FVC) is the percentage of the area of canopy and branches of all vegetation (including forest, shrub land, grassland, and arable land) to the total area, and ranges from 0 to 1. It is the most important parameter for monitoring the ecological environment [84]. When a pixel of a remote sensing image is completely filled with geohazards (i.e., almost no vegetation coverage remains), the value of *FVC* is close to 0; when a pixel is completely covered by vegetation (i.e., almost no geohazards have affected the area), the value is close to 1; and when a pixel is a mixture of geohazards and vegetation coverage, the value is >0 and <1. Thus, the higher the degree of vegetation coverage, the closer the *FVC* is to unity. Based on NDVI, the dimidiate pixel model is currently the most widely used method for retrieving *FVC* [85].

The calculation of *FVC* with NDVI is based on the assumption that the contributions to a pixel value consist of vegetation and soil [84,86,87]. If the vegetation coverage is $f_c$, then the soil coverage is $1 - f_c$, and *NDVI* can be approximately expressed as

$$NDVI = f_c \times NDVI_{veg} + (1 - f_c) \times NDVI_{soil} \tag{1}$$

Thus, *FVC* can be expressed as

$$f_c = \frac{NDVI - NDVI_{soil}}{NDVI_{veg} - NDVI_{soil}} \tag{2}$$

Here, $f_c$ represents the *FVC*; *NDVI* is the value of MODIS-NDVI pixel; and $NDVI_{soil}$ represents the *NDVI* value of pixels that are bare soil. In this study, the 5% of pixels with the lowest *NDVI* values correspond to areas of bare soil, and the ratio is the number of the bare soil *NDVI* pixels to the total number of *NDVI* pixels in the study area. The $NDVI_{soil}$ can be empirically determined by the lower limit of the bare soil pixels corresponding to the *NDVI* values. The $NDVI_{veg}$ represents the *NDVI* value of pixels that are pure vegetation. The 5% of pixels with highest *NDVI* values corresponding to the areas entire covered by vegetation, and the ratio is the number of purely vegetated *NDVI* pixels to the total number of *NDVI* pixels in the study area. The $NDVI_{veg}$ can be empirically determined by the upper limit value of the pure vegetation pixels corresponding to the *NDVI* values.

Based on the fitted MODIS-NDVI time series, *FVC* of vegetation growing season (May to September) was calculated using Equation (2). The MVC method was then used to calculate the maximum value of *FVC* in the growing season for each year. The 16 *FVC* were thus obtained for 2001–2016.

### 2.3.4. Extraction of the Proportion of Damaged Vegetation

Based on the assumption that the vegetation damage rate (VDR) was consistent with the rate of occurrence of geohazards, the *VDR* was extracted from *FVC* using Equation (3) [50,51,88].

$$VDR_{i+1} = \frac{FVC_i - FVC_{i+1}}{FVC_i} \times 100\% \tag{3}$$

Here, $VDR_{i+1} > 0$ indicates that the *FVC* value in year $i + 1$ is lower than $i$, and $i$ represents the year from 2001 to 2016. In principle, the occurrence of geohazards leads to a decrease of *FVC*. This enabled data for $VDR_{2002}, \ldots, VDR_{2008}, VDR_{2009}, VDR_{2011}, VDR_{2013}, \ldots, VDR_{2016}$ to be produced. Since there was a large amount of cloud coverage in the study area during 2010 and 2012 (based on the reliability band data for MOD13Q1, a value of 3 indicates that the observation targets were obscured by clouds or were poorly visible), the data for $VDR_{2010}$ and $VDR_{2012}$ were not calculated. $VDR_{2011}$ and $VDR_{2013}$ were calculated using Equations (4) and (5), below.

$$VDR_{2011} = \frac{FVC_{2009} - FVC_{2011}}{FVC_{2009}} \times 100\% \tag{4}$$

$$VDR_{2013} = \frac{FVC_{2011} - FVC_{2013}}{FVC_{2011}} \times 100\% \tag{5}$$

### 2.3.5. Calculation of the Area of Damaged Vegetation

The empirical threshold method has previously been used to estimate the AVD caused by co-seismic geohazards based on NDVI data before and after the 12 May 2008 Wenchuan Earthquake [89–91]. However, the areas of geohazard occurrence are spatially heterogeneous and the empirical threshold method is unsuitable in such a complex geological environment, especially in areas with varying degrees of vegetation destruction. A segmented logistical function has also been used to characterize annual vegetation dynamics by calculating the maximum and minimum value of changes in curvature [92,93]. Another approach has combined the use of the two methods to construct the curvature function of VDR using Equation (6) (see Appendix A for a detailed derivation of this equation), below [50]. The AVD is obtained by setting the quantitative threshold to the maximum value of the changing curvature. The curvature function of the VDR histogram is calculated as follows:

$$Curvature(VDR) = \frac{\left| f''(VDR) \right|}{\left( 1 + f'^2(VDR) \right)^{\frac{3}{2}}} = \frac{\left| f(VDR + 2) + f(VDR) - 2f(VDR + 1) \right|}{\left( 1 + (f(VDR + 1) - f(VDR))^2 \right)^{\frac{3}{2}}} \tag{6}$$

Here, $Curvature(VDR)$ is the curvature function of the VDR histogram, and $f(VDR)$ is the histogram function of *FVC*. Comparison with $k(k > 0)$ multiply the standard deviation ($\sigma$) and the maximum curvature function of $Curvature(VDR)$ indicates that most of the AVD is located within $3\sigma$. The maximum of the curvature function for the histogram can be measured by the open interval $\left( \overline{VDR}, \min\left\{ \overline{VDR} + 3\sigma, VDR_{max} \right\} \right), k = 3$; if the maximum value of VDR of the corresponding curvature is the required threshold value $T\left( T = \{curvature(VDR)\}_{max}, VDR \in \left( \overline{VDR}, \min\left\{ \overline{VDR} + 3\sigma, VDR_{max} \right\}, k = 3 \right) \right)$, then the AVD will be obtained by the threshold value $T$.

### 2.3.6. Verifying the AVD

The AVD calculated by step (5) in Figure 2 requires further field surveys or other RSI verification to eliminate the influence of "non-geo-hazard" factors, including seasonal changes in vegetation and engineering activity, which result in a decrease in NDVI values.

To eliminate the effects of factors such as variations in the data quality of the RSI and geohazard attributes (e.g., the number, area, scale and spatial location in different years), multi-source data were used to verify the effectiveness of the extracted results. In addition, the consistency of the extracted results with other data from various sources was evaluated on both small and large scales. The detailed procedure used was as follows:

(1)　For a small-scale area, near the epicenter, the results of geohazards (using new landslides and debris flows of all scales) [74,75] were overlapped with the extracted AVD (caused by large and medium-scale geohazards), and were used to verify the reliability of our approach. For

the Gengda to Yingxiu section of Province Road (PR303) in Wenchuan County, with an area of ~85 km$^2$, debris flows were triggered after three heavy rainfall events (on 14 August 2010; 4 July 2011; 10 July 2013). The total number and area of new landslides and debris flows in 2008, 2010, 2011 and 2013 were interpreted from high spatial-resolution images (QuickBird, 0.61 m; Worldwiew-2, 2.5 m; Rapid-Eye, 5 m; SPOT-6, 0.5 m). The total volume of hillslope and channelized deposits was measured by field surveys using an in situ laser range finder.

(2)　For a large-scale area, in order to analyze the spatiotemporal distribution of the decreasing trend of LMG during 2001–2016, we used the same methodology to extract the AVD and compared our findings with other RSI (e.g., Landsat, Gao-Fen and QuickBird) and the geohazard points of local reports produced by the senior geohazards management agency. Three time-phases $(i, i+1, i+2, i = 2001, \ldots\ldots 2016)$ of RSI before and after the occurrence of geohazards were selected to evaluate the effectiveness of the method (Figure 3).

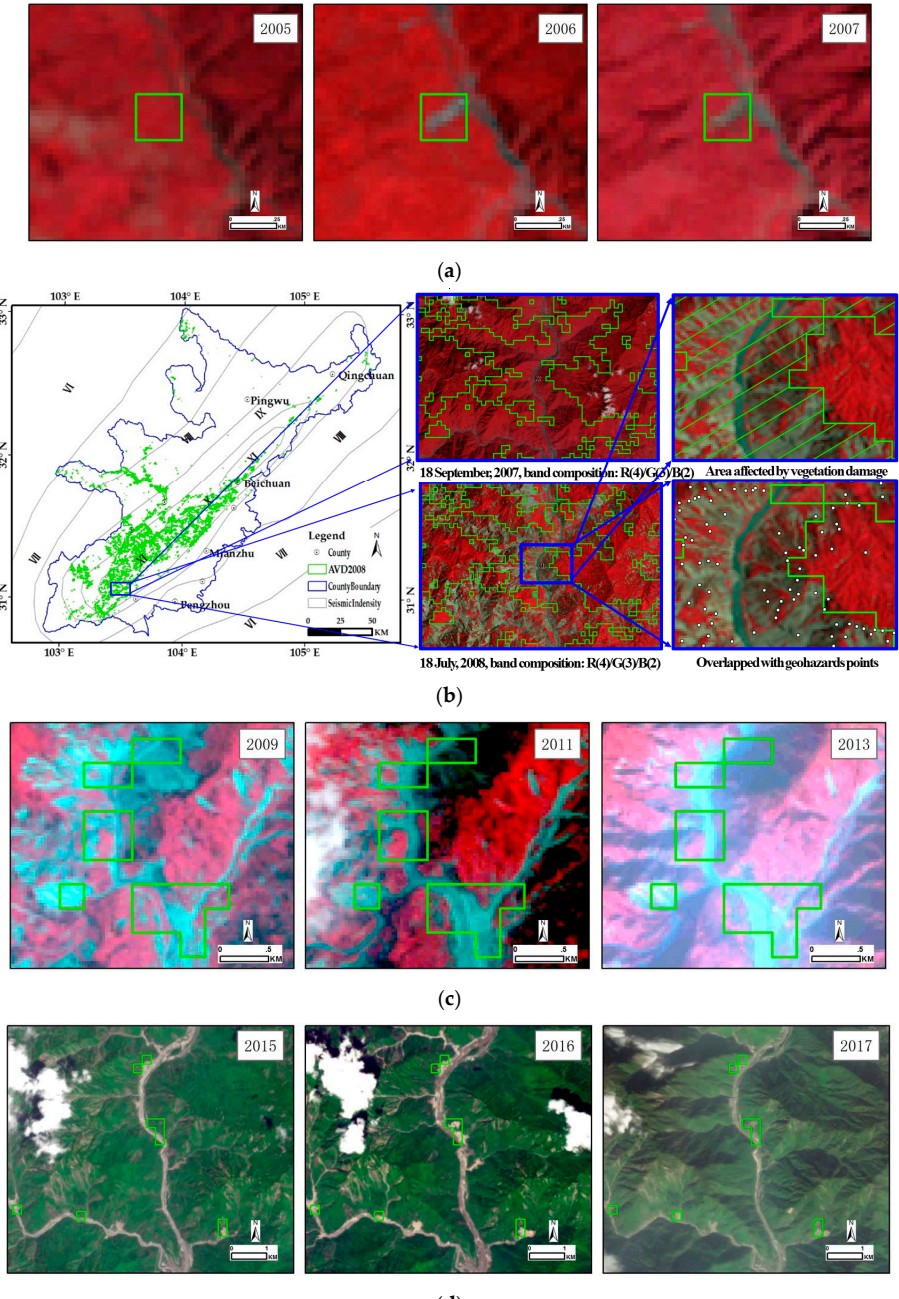

**Figure 3.** Verification of the extracted area affected by vegetation damage (AVD), caused by geohazards, for study area for 2006, 2008, 2011, and 2016. The green polygon of the $i+1$ year represents the extracted AVD by MODIS. The $i$ year RSI are used as a reference background information in areas where no geohazards have occurred by comparing with the $i+1$ year. The $i+2$ year RSI are used to corroborate the extracted AVD results of the $i+1$ year based on the fact that these areas will not be quickly covered by vegetation in the following year after the occurrence of geohazards. (**a**) For Wenchuan County, the green vector polygon represents the extracted AVD for 2006 (caused by geohazards). The RSI on the left, middle and right show the same place and are from Landsat-5 with a spatial resolution of 30 m. Path/Row is 130/038; band composite is R(4)/G(3)/B(2); and the data acquisition dates are 26 July 2005, 27 June 2006, and 6 May 2007, respectively. From the visual interpretation of Landsat-5, red coloration represents the areas covered by vegetation. The bright gray coloration represents geohazards such as landslides and collapses. (**b**) In the map on the left, the green vector polygon represents the extracted AVD for 2008 (caused by co-seismic geohazards) and the blue rectangle shows the location of the RSI on the right. In the RSI on the right, the two leftmost RSI are from Landsat-5 with a spatial resolution of 30 m in locality A (see Figure 1 for location). Path/Row is 130/039; band composite is R(4)/G(3)/B(2); and the data acquisition dates are 18 September 2007 (top) and 18 July 2008 (bottom). The blue rectangle in the lower image shows the location of the enlarged area shown in the two rightmost images. In the lower of the two rightmost images, the white dots are geohazards revealed by actual field surveys (the expiration date is 31 December 2008). From visual interpretation, the white coloration in the left of middle image are clouds. (**c**) Locality C (see Figure 1 for location). The left and middle RSI are from Landsat-5 with a spatial resolution of 30 m. Path/Row is 130/038; band composite is R(4)/G(3)/B(2); and data acquisition dates are 3 June 2009 and 28 August 2011, respectively. The RSI on the right is from Landsat-8 with a spatial resolution of 30 m. Path/Row is 130/038; band composite is R(5)/G(4)/B(3); and the data acquisition date is 7 December 2013. The RSI on the left, middle and right show the same place. From the visual interpretation of Landsat-5 and Landsat-8, red coloration represents the areas covered by vegetation. The bright gray-blue coloration represents geohazards such as landslides, collapses and debris flows. (**d**) Locality A (see Figure 1 for location). The RSI on the left, middle and right show the same place and are from Gao-Fen 1 with a spatial resolution of 2 m. Band composite is R(3)/G(2)/B(1); data acquisition dates are August 2015, August 2016 and November 2017, respectively. From visual interpretation, the white coloration in these three images are clouds. The green coloration represents the areas covered by vegetation. The bright gray coloration represents geohazards such as landslides, collapses and debris flows.

## 3. Results

### 3.1. FVC before the 12 May Wenchuan Earthquake

Using the methodology described above, calculation of the NDVI values from the Landsat-5 images for June–September, 2007 (i.e., before the Wenchuan Earthquake) revealed that the FVC of the 10 counties/cities reached 92%.

### 3.2. Fitted NDVI

The fitted NDVI were effectively eliminated by the cause of background noise. The fitted NDVI confirmed previous findings [38,46,94,95] that co-seismic geohazards associated with the 12 May 2008 Wenchuan Earthquake led to a decrease in NDVI values. Although the fitted NDVI value is not the true value, it nevertheless reflects changes in the trend of NDVI for the entire curve of the vegetation for all status in different localities of NDVI pixels with three main characteristics:

(i)    The vegetation was not affected by co-seismic geohazards either before or after the 12 May 2008 Wenchuan Earthquake. The maximum NDVI value of the pixel-level (corresponding to the vegetation growing season, from May to September) was relatively uniform during 2001–2016 (Figure 4).

(ii)　The vegetation was seriously damaged by the 12 May 2008 Wenchuan Earthquake. Compared with the preceding year (2007), the maximum NDVI value decreased substantially in 2008 but rose continuously during 2009–2016 (Figure 5). Alternatively, the maximum NDVI value did not change much during 2009–2016, following the steep decline in 2008, and was consistent with the maximum value in 2008. A third alternative was that the maximum NDVI values decreased sharply in 2008, then increased gradually, and then decreased sharply again in subsequent years and remained at a low level.

(iii)　The vegetation was unaffected or only slightly affected by the 12 May 2008 Wenchuan Earthquake. However, in the years after 2008, the vegetation was damaged by geohazards, and there was a decrease in the maximum value of $NDVI_i$ ($i \geq 2009$) (Figure 6). With time, the evolution of damaged vegetation exhibited patterns of rapid recovery, slow recovery, or renewed destruction following restoration. The maximum value of $NDVI_i$ ($i \geq 2009$) corresponded to a rapid rise, slow rise, or rapid decline.

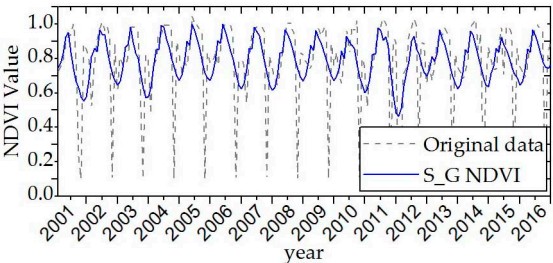

**Figure 4.** The minimal change in fitted values of an NDVI (Normalized Difference Vegetation Index) pixel in time series near the Huojizai Village of Maoxian County before and after the 12 May 2008 Wenchuan Earthquake. In Figures 4–6, the broken grey lines are the original MODIS-NDVI data and the solid blue lines are the fitted data.

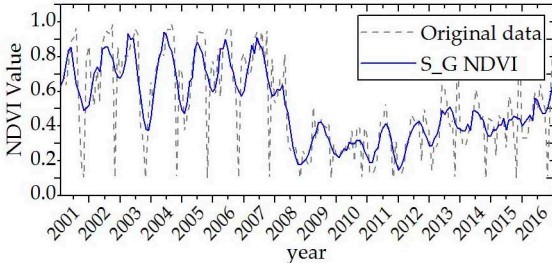

**Figure 5.** The maximum value of a fitted NDVI pixel decreased sharply in 2008 on the Daguangbao landslide in Anxian County, as a result of the 12 May Wenchuan Earthquake, and dropped again in 2009 and 2010, then increased gradually in the following years.

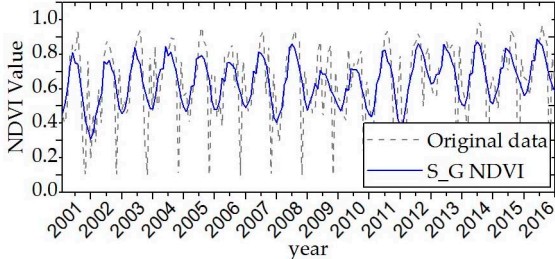

**Figure 6.** The maximum value of a fitted NDVI pixel decreased slightly in 2009 and then increased gradually, which is located in the Xibeiqiao Village of Pengzhou City. This reflects the minor influence of the 12 May 2008 Wenchuan Earthquake.

*3.3. Validity of the Methodology*

At a small scale, assuming that the results and field surveys in 2008, 2010, 2011 and 2013 [74] are reliable, we found that the degree of consistency for 2008 with the extracted AVD in the proposed method was 98.02%, for 2011 it was 80.26%, and for 2013 it was 78%. We did not extract the AVD in 2010 and therefore the degree of consistency for 2010 could not be evaluated.

For the whole study area of 10 Counties/Cities, there were 11,155 co-seismic geohazard sites based on the extracted AVD of 2008 [72]; the degree of consistency was 98.30%. In addition, the area of extracted AVD for 2008 was ~40 km$^2$ larger than was previously reported [70]. However, our results are consistent with previous estimates of ~1249 km$^2$ [39] and ~1159 km$^2$ [68]. For the interval prior to the 12 May 2008 Wenchuan Earthquake (2001–2007), the total number of geohazards reported by a local management agency was 135. This estimate was based on a survey which was confined to accessible areas. The survey was mainly small-scale and was unsuitable for evaluating the degree of consistency with the extracted AVD before the 12 May 2008 Wenchuan Earthquake. During the post-earthquake period (2009–2016), the number of geohazard sites determined by field surveys conducted by the Sichuan environmental monitoring station was 3307, among which 113 were large and huge-scale with a volume greater than 100,000 m$^3$. The extracted AVD produced by the method proposed here from MODIS-NDVI time series covered 95 of the total of 113 large and huge-scale geohazard sites, demonstrating an 84.07% degree of consistency between the field survey results and the digitally extracted results from the RSI.

*3.4. Area of Damaged Vegetation Caused by LMG*

The maps of the spatial distribution of AVD caused by geohazards for the period prior to the 12 May 2008 Wenchuan Earthquake (2001–2007) (Figure 7a–f) reveal the geohazards are discrete and relatively well-separated which also demonstrates that the geological environment was relatively stable before the 12 May 2008 Wenchuan Earthquake, and was mainly concentrated in mountain valley areas with steep terrain, along the valleys of the Minjiang River and Jianjiang River and in the Zagunao watershed. These results are consistent with previous research [69]. Table 1 lists the areas of AVD for the study area of Wenchuan County and localities A–E, from 2001 to 2016.

The distribution of AVD in 2008 is consistent with the orientation of the Longmenshan Fault Zone, which extends linearly from southwest to northeast (Figure 7g). The co-seismic geohazards were large in terms of number and scale, and exhibited a centralized and continuous distribution, while the area of AVD was hundreds of times larger than that prior to the earthquake. The closer to the epicenter, the greater the intensity of vegetation damage. In regions with high seismic intensity, the AVD had a cascading distribution and was highly consistent with the scale of geohazards.

The maps for the period after the 12 May 2008 Wenchuan Earthquake (2009–2016) (Figure 7h–n) show that the AVD, reflecting the occurrence of geohazards, occurred not only on the original unstable slopes, but also in new areas. For the whole research area, the area of AVD during 2009–2016 exhibits a decreasing trend; however, it was still several times larger than before the 12 May 2008 Wenchuan Earthquake, which implies that a high intensity of geohazards activity was maintained for at least eight years and that the geological environment remained unstable.

**Table 1.** Area affected by vegetation damage (AVD) of the study area.

| Study Area | Total Area (km$^2$) | Total Cumulative Area of AVD (2001–2007) (km$^2$) | Area of AVD (2008) (km$^2$) | Total Cumulative Area of AVD (2009–2016) (km$^2$) |
|---|---|---|---|---|
| Study area | 25,813 | 19 | 1719 | 544 |
| Wenchuan | 4093 | 4 | 690 | 106 |
| A | 336 | 3 | 293 | 25 |
| B | 569 | 0.19 | 493 | 38 |
| C | 1217 | 0.77 | 599 | 73 |
| D | 988 | 0 | 60 | 21 |
| E | 785 | 0 | 13 | 7 |

The AVD during 2009–2016 also exhibits a decreasing trend in terms of spatial location along the fault zones from the southwest to the northeast. Among the five localities, geohazard activities were more frequent in localities B (Hongkou-Yinchanggou) and C (Hongbai-Chaping), which were located in seismic intensity zone XI, than elsewhere. Although the overall trend of AVD was decreasing, the area of AVD was still several times higher than that before the 12 May 2008 Wenchuan Earthquake. The total area of extracted AVD for the 10 Counties/Cities before the 12 May 2008 Wenchuan Earthquake (2001–2007), during the co-seismic period (2008), and during the post-earthquake period (2009–2016), were 19 km$^2$, 1719 km$^2$ and 544 km$^2$, respectively.

A graph of the total area of vegetation damaged by geohazards for the 10 Counties/Cities from 2002–2016 is shown in Figure 8. The "post-earthquake effect" of the geohazards is obvious. The geohazards are mainly concentrated within the interval of 2009–2013 and they peaked in 2011. During 2014–2016 the area of AVD commenced a decreasing trend, but it was still larger than that before the 12 May 2008 Wenchuan Earthquake. The total area of geohazards in 2014 was three times that before the earthquake (2001–2007). These results are consistent with previous findings [25,77], which suggested that the areas of vegetation damage would maintain a high level of geohazard activity for five years following the 12 May 2008 Wenchuan Earthquake (Figure 9) [25].

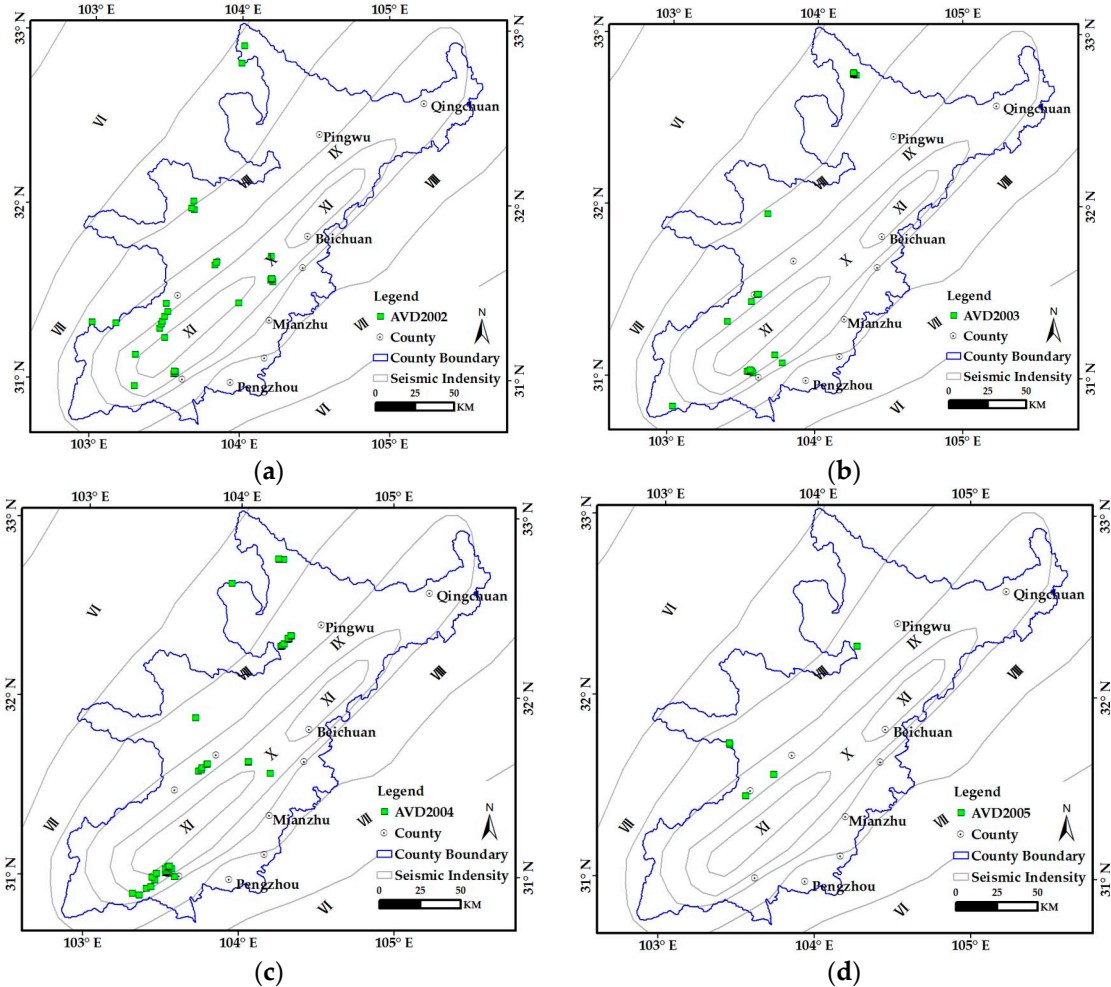

**Figure 7.** *Cont.*

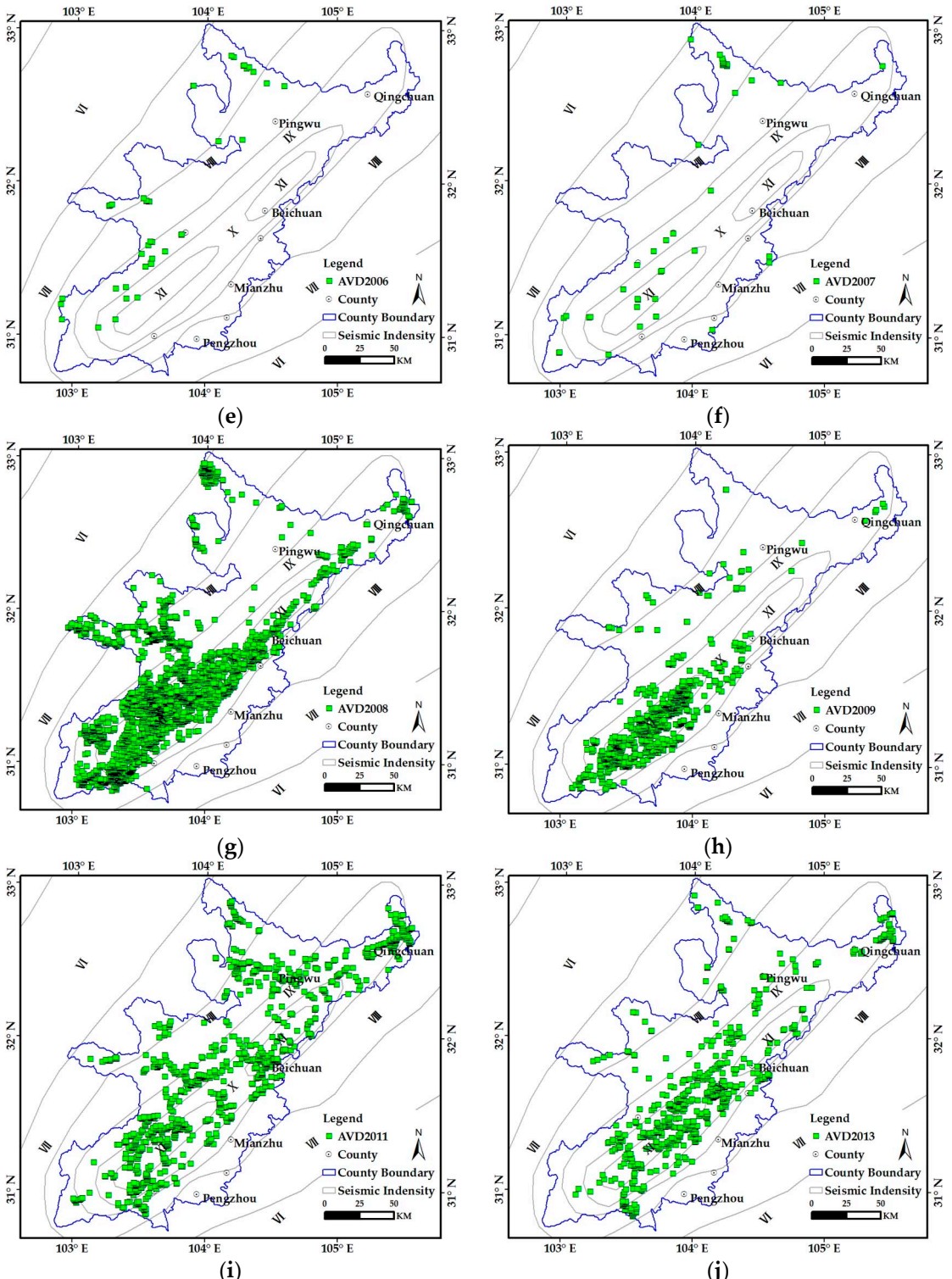

**Figure 7.** *Cont.*

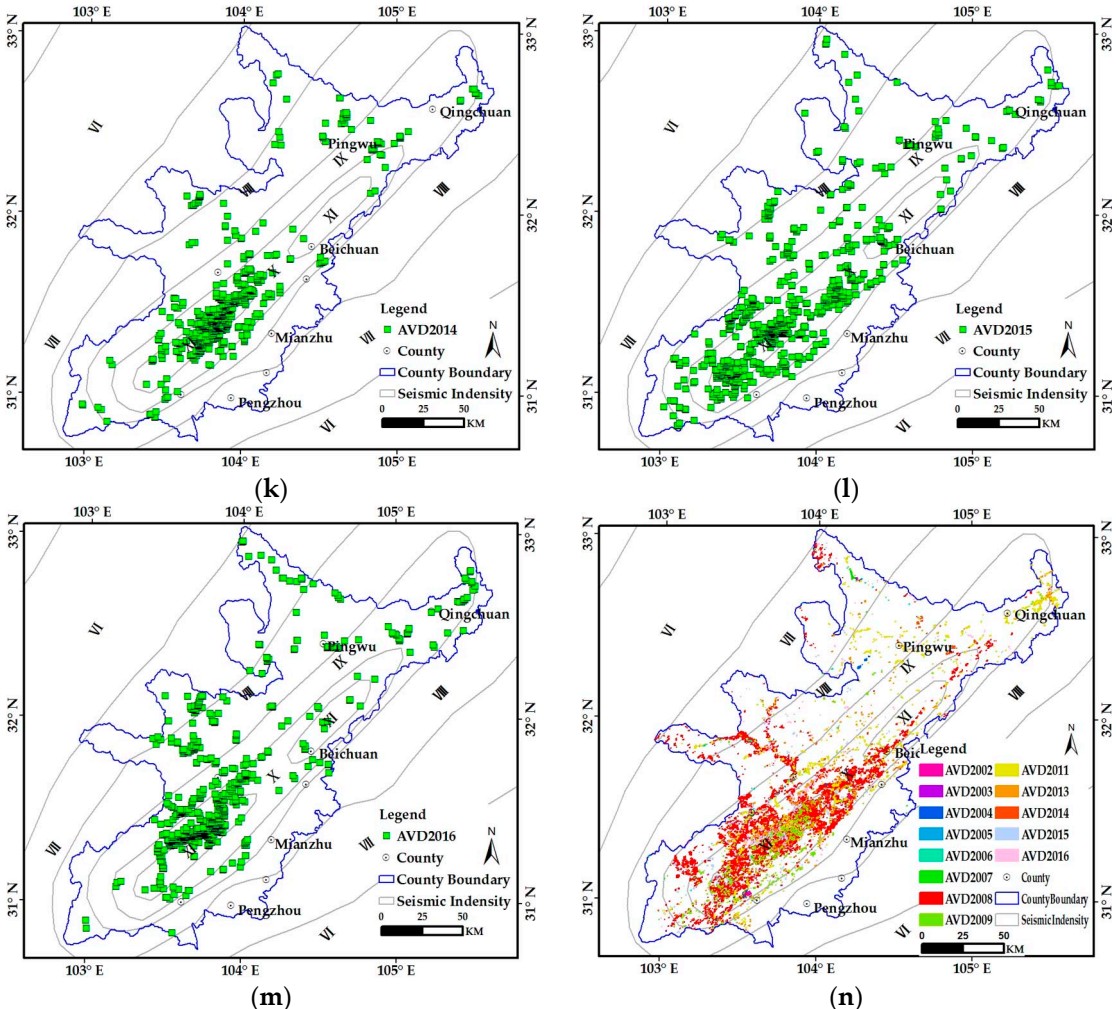

**Figure 7.** Distribution of the areas of vegetation damage caused by geohazards during 2002–2016. Maps (**a**–**m**) represent the AVD for 2002, 2003, 2004, 2005, 2006, 2007, 2008, 2009, 2011, 2013, 2014, 2015, 2016, respectively. The green squares show the location of geohazard points or groups. In reality, the extracted AVD is a polygon, but for clarity the AVD locations are shown by green squares. The black dots represent County/City locations. The blue polygon represents the study area, and the gray polygons represent seismic intensity levels from VI–XI. (**n**). Distribution of AVD caused by geohazards for the entire period of 2002–2016.

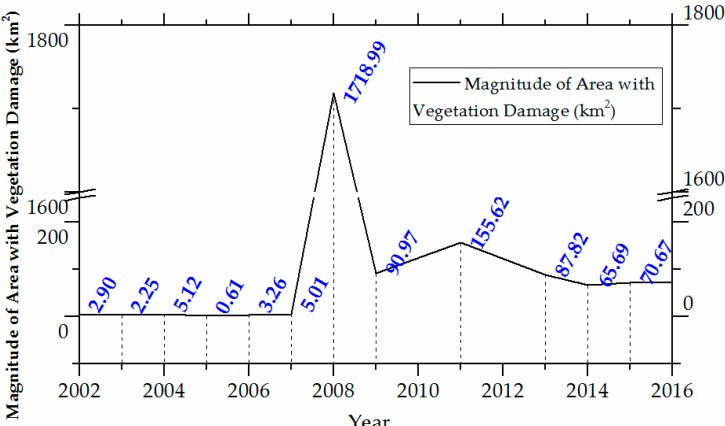

**Figure 8.** Change in AVD caused by geohazards in the study area of 10 Counties/Cities from 2002 to 2016.

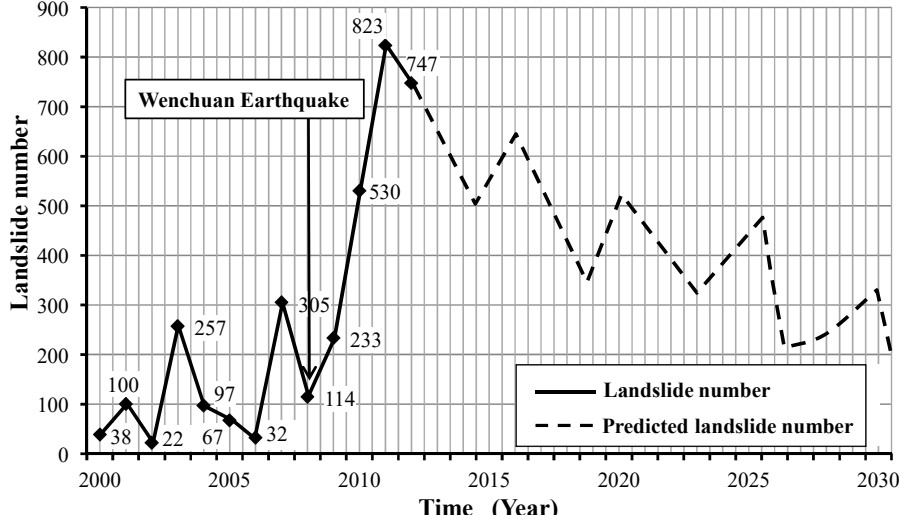

**Figure 9.** Incidence of catastrophic landslides induced by rainfall in the Wenchuan Earthquake area since 2000. Only landslides that caused losses to life and property are considered. The solid line is based on the actual records and the dashed line is a prediction [25].

### 3.5. Relationship between AVD and the 12 May 2008 Wenchuan Earthquake

The results for the entire study area, including Wenchuan County and the five localities, reveal the following:

(i)   Before the earthquake (2001–2007), the total cumulative area of AVD at the three spatial scales studied herein (i.e., the entire study area, Wenchuan county, and the five localities) was small (Figures 8 and 10). According to the USGS statistics (Figure 11), the number of earthquakes of magnitude Mw > 3 in the region during 2001–2007 was small. This indicates that the AVD was not directly positively correlated with the occurrence of the 12 May Wenchuan Earthquake. From the previous results, geohazards are mainly affected by rainfall, topography and lithology.

(ii)  The 12 May 2008 Wenchuan Earthquake triggered a large number of geohazards which resulted in a total area of AVD at the three spatial scales that was 2-3 orders of magnitude greater than before the earthquake (during 2001–2007). According to USGS statistics (Figure 11), the number of earthquakes in 2008 was also very large. From Figures 10 and 11, which respectively show time series of AVD and the number of earthquakes for the study region from 2002–2016, it can be seen that there is a strong positive relationship between AVD and the number of earthquakes in 2008. The magnitude of the earthquakes is concentrated within the range of Mw 4-Mw 4.9, which implies that the AVD was related to the number of shallow earthquakes (depth < 10 km) (Figure 12).

(iii) After the 12 May 2008 Wenchuan Earthquake (2009–2016), the area of AVD at the three spatial scales began to decrease gradually, but with substantial fluctuations. According to the USGS statistics (Figure 11), the number of earthquakes generally decreased after 2008, and the magnitude was reduced; however, for several individual years the number of earthquakes was still large, as was the AVD. This indicates a positive relationship between the AVD and the number of earthquakes during 2009–2016, and that the geological environment remained unstable, especially in specific locations.

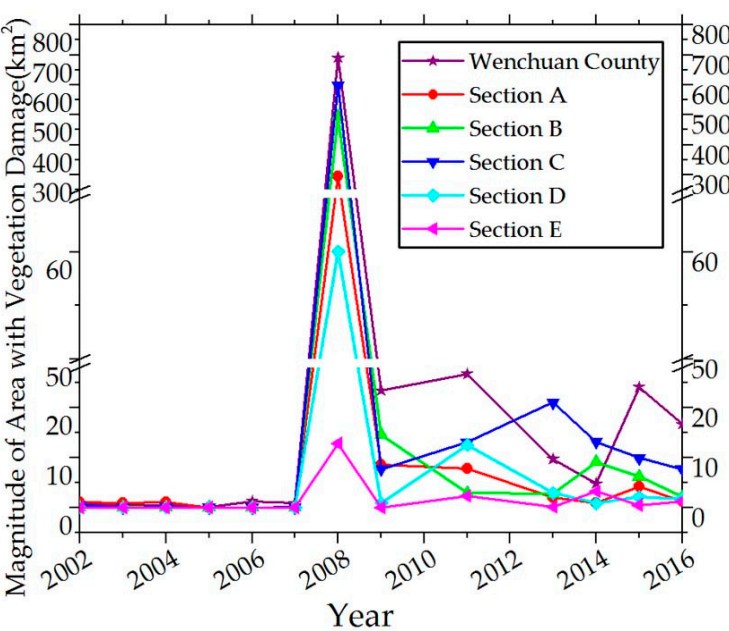

**Figure 10.** Time series of AVD in Wenchuan County and the five localities from 2002 to 2016.

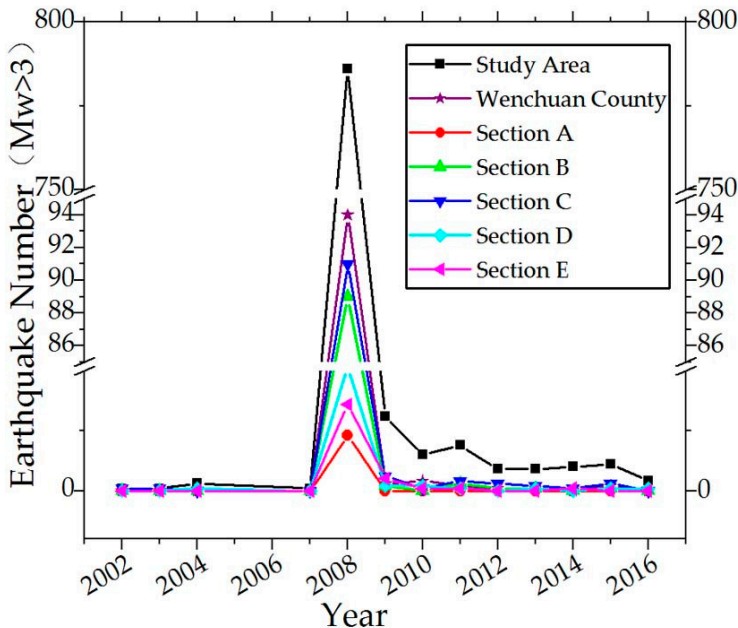

**Figure 11.** Time series of the annual incidence of earthquakes of Mw >3 in the study area from 2002 to 2016 (Source: USGS).

The area of AVD caused by post-earthquake geohazards varied between years (Figure 10), which implies that the frequency of geohazard activity in the five localities remained high. For Wenchuan County, the maximum area of AVD was in 2011, following two consecutive years of decrease in 2013 and 2014; the AVD peaked again in 2015 but did not exceed the value for 2011. For locality A, the area of AVD indicates two small peak in geohazards in 2011 and 2015, but the area of AVD in 2015 was less than that in 2011. For locality B, the vegetation cover (reflected by the AVD) was severely damaged by the 12 May 2008 Wenchuan Earthquake, which was also related to the number of shallow earthquakes which occurred in 2008 (Figure 12). For the five consecutive years from 2009–2013, the area of AVD decreased continuously, but after the peak in 2014 it began to decrease again. For locality C, the vegetation cover was also severely damaged by the 12 May 2008 Wenchuan Earthquake, which

was also related to the number of shallow earthquakes in that year (Figure 12); in addition, the area of AVD increased from 2009–2013 and then decreased from 2014 onwards. For locality D, the area of AVD exhibits two small peaks (2011 and 2014) but the area of AVD in 2014 did not exceed that in 2011. For locality E, the area of AVD exhibits two small peaks (in 2011 and 2014) of very similar magnitude.

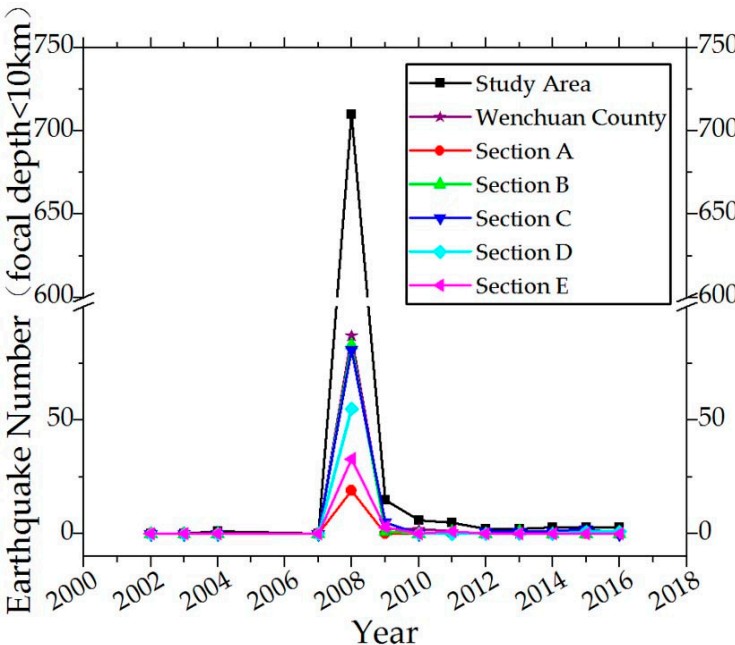

**Figure 12.** Time series of the number of earthquakes (depth < 10 km) in the study area from 2002 to 2016 (source: USGS).

After the Wenchuan Earthquake (2009–2016) the areas of AVD for localities B and C are larger than those of the other localities (Figure 10). A close relationship remained between AVD and seismic intensity level XI and the seismogenic fault, which was related to the high incidence of earthquakes in localities B and C following the 12 May 2008 Wenchuan Earthquake (Figure 11); the magnitudes were mainly in the range of Mw 4-Mw 4.9 (Figure 13). In addition, in contrast with localities A, D, and E, locality C is located in the watershed of the Mianyuan River, which was repeatedly affected by debris flows following the 12 May 2008 Wenchuan Earthquake; for example the Qingping debris flows occurred on 13 August 2010 and 19 August 2010-, and the Gaochuan debris flows occurred on 17 August 2009. Locality B is located in the watershed of Baisha River, where the Yinchang gully debris flows occurred on 18 August 2012. Our findings are consistent with previous research [22,25,76,96,97]. Due to the limitations of the precipitation data, there was only one dataset, from Dujiangyan Observatory Station (elevation 698.5 m) for the interval from January, 2001 to December, 2016; this dataset shows that precipitation was mainly concentrated during June-September before the 12 May 2008 Wenchuan Earthquake, as was the case for many years previously (Figure 14). However, following the 12 May 2008 Wenchuan Earthquake, the total cumulative precipitation peaked in 2013, which enables us to analyze the relationship between the occurrence of geohazards and the minimum rainfall threshold in section B.

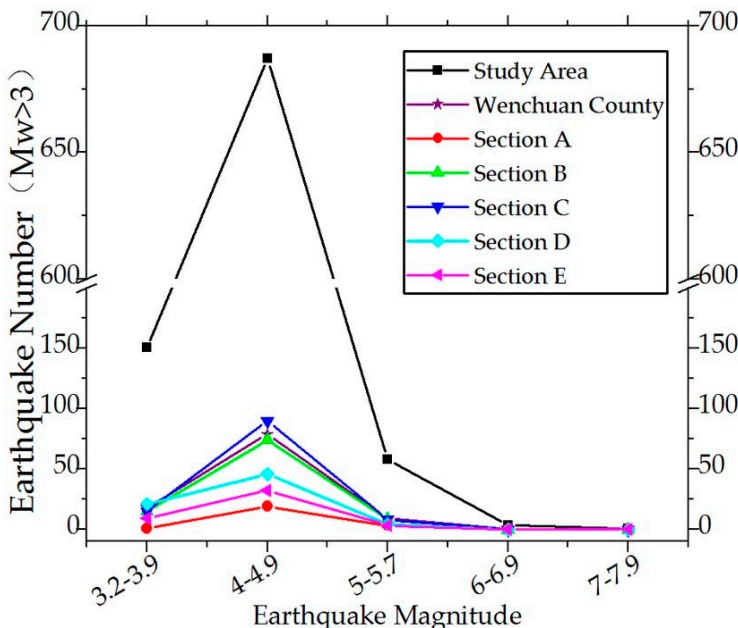

**Figure 13.** Relationship between earthquake magnitude (Mw > 3) and earthquake number in the study area from 2002 to 2016 (source: USGS).

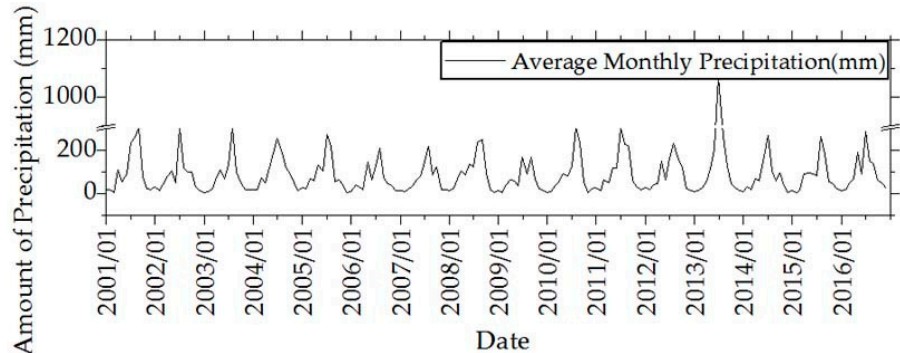

**Figure 14.** Average monthly precipitation (mm) at Dujiangyan Meterological Observatory from 2001 to 2016.

## 4. Discussion

### 4.1. FVC before the Wenchuan Earthquake

Our results of FVC before the Wenchuan Earthquake (92%) are consistent with those of other workers [54], who used land cover data of MODIS product (MOD12Q1, 1 km spatial resolution) before the 12 May 2008 Wenchuan Earthquake to estimate that more than 98% of the land surface in the Wenchuan region (a large area comprising more than 10 counties/cities) was densely vegetated. Another study [38] estimated that the FVC of Pingwu County was 96.6% in 2000; in addition, it was observed that there was little overlap between areas of different land use (e.g., from forest to farmland), and that an area affected by landslides was present. The land use was interpreted using two Landsat-5 images, from 3 May 2000 and 19 May 2006.

These results indicate that the study area was densely vegetated prior to the 2008 Wenchuan earthquake, and they provide the basis for quantifying the occurrence of geohazards based on the observed vegetation damage. The following points are also noteworthy:

(i)     In theory, as a result of the spatial resolution of MOD13Q1 (250 m), small-scale geohazards could not be detected. In other words, where the ruptured surface region caused by geohazards is

much smaller than one pixel of the MODIS-NDVI (62,500 m$^2$), geohazards could not be detected. However, if the distance of landslide movement is sufficient, even if they are of small-scale (volume), the landslides could be detected as long as they occurred within areas with a high degree of vegetation coverage prior to the occurrence of geohazards.

(ii)   Under the influence of rainfall, a creeping landslide would be formed by the loose materials on unstable slopes. This type of landslide does not cause vegetation damage and therefore it could not be detected.

## 4.2. Reliability of the Method

Monitoring of the area and analysis of the spatial distribution of geohazards was conducted in 2008 by rapidly extracting the AVD using a threshold value of differentiated NDVI datasets (MOD09Q1, 250 m spatial resolution, taken on 14 April 2008 to 17 June 2008) before and after the 12 May 2008 Wenchuan Earthquake [54]; the analysis was facilitated by the high degree of vegetation coverage of the area prior to the earthquake. The results were interpreted using ASTER images with a long temporal duration (15 m spatial resolution, 19 February 2003 and 23 May 2008) and were used to determine the actual areas of all scales of geohazards caused by the 12 May 2008 Wenchuan Earthquake. The 54% of the landslides were detected and 20.4% of detected 'landslides' were in fact non-landslides; in addition, 25.2% of landslides were not detected. Notably, during the interval of 2003–2008 (the dates of the ASTER images), besides the occurrence of geohazards, there were also changes in land use, which led to a relatively larger reference value and a smaller comparison value. In other words, the extracted AVD (250 m spatial resolution) only corresponded to geohazards of large and medium-scale. This was the main reason for the low degree of consistency. In addition, it is possible that the spatial occurrence of geohazards obtained using the simple threshold method may have been underestimated due to the failure to take into account the effects of noise (i.e., interference from clouds). Consequently, a new method of setting the threshold was proposed [46]. This used an iterative search process to determine the optimal window length, which was used to extract the AVD from MOD09Q1 (8-day composite products of MODIS-NDVI with 250 m spatial resolution, from 29 February 2008 to 17 June 2008). The consistency of the occurrence of geohazards has been improved by comparing the interpretation results of landslides from two SPOT-5 images (5 September 2006 and 4 June 2008) of an area of 562 km$^2$, the success rate for identification reached 75%. Therefore, these authors further suggested that this approach using high spatial resolution time-series NDVI could potentially identify landslides and the subsequent triggering processes. After constructing pre-earthquake benchmarks (MOD09Q1, 2000 to 2007), the AVD caused by co-seismic landslides was detected by comparing all NDVI bands of the post-earthquake period (2008 to 2012) to the benchmarks; this was then used as the basis for proposals for the restoration of the damaged vegetation [38]. The validity of the method was verified with reference to the co-seismic landslides inventory, which was interpreted using SPOT-5 images and field work.

In this study, we propose that the monitoring of geohazards can be conducted by detecting the area of damaged vegetation. The method was applicable to the study area because of the high degree of vegetation coverage before the occurrence of geohazards. The foregoing indicates that AVD has a high positive correlation with the occurrence of co-seismic geo-hazards. The fitting method was adopted to extract the AVD by eliminating as many types of interference as possible, and the improved threshold method can maximize the degree of consistency. In a small region, the consistency was higher in 2008 and then gradually decreased during 2009–2016, which was related to the scale of geohazards and the spatial resolution of RSI.

By this paper's extracted method, the small parts of AVD are misidentified by their curvatures are not significant in the middle of AVD due to some small-scale geohazards. Although this paper pays more attention to the law of the occurrence of large and medium-scale geohazards, these small geohazards are also worthy to be paid attention to, especially when the frequency of small-scale geohazards is high in local areas. In order to reduce these misidentifications, this paper verifies the

extraction results by comparing the visual interpretation results from high spatial resolution remote sensing images. In addition, because of cloud interference, rainfall and topographical shadows, information in some areas was missing, which made it impossible to establish NDVI time-series with a spatial resolution higher than 30 m at the regional level (10 counties/cities). According to statistics from the Geospatial Data Cloud website (http://www.gscloud.cn/), and using the data of the Landsat satellite series as an example (Landsat-5 retired on 5 June 2013), for the 10 Counties/Cities, 6 scenes were needed with Path/Row of 129/037, 130/037, 129/038, 130/038, 129/039, and 130/039. Influenced by the weather, only 25% of the total data could be obtained in the region, and only for 13% of the total data was the cloud coverage less than 10%. Landsat-7 was launched on 15 April 1999, but the onboard scanning line corrector (SLC) malfunctioned on 31 May 2003, which affected the availability of ETM+ by data strip loss. Landsat-8 was launched on 11 February 2013. Although it provides high spatial resolution RSI (Aerial photos, 2 m; Quick-Bird, 0.61 m; IKONOS, 1 m; SPOT, 2.5 m) which can be used for fusion or interpolation to complete the construction of NDVI time series for a small watershed, the economic costs and time required are excessive. In addition to changes in land use and extreme climate events that would lead to changes in NDVI, the effects of uneven haze on NDVI should also be considered. Moreover, the complex terrain of the study area would increase the BRDF (Bidirectional Reflectance Distribution Function) effect, which should also be considered.

## 5. Conclusions

The 12 May 2008 Wenchuan Earthquake triggered a very large number of co-seismic geohazards, and the aftershocks, combined with the effects of rainfall, resulted in the frequent occurrence of secondary geohazards. With the gradual reduction in the amount of unconsolidated material, there was a decreasing temporal trend of geohazards within the areas affected by the 12 May 2008 Wenchuan Earthquake. The co-seismic and secondary geohazards after the 12 May 2008 Wenchuan Earthquake seriously damaged the vegetation cover, and the NDVI time series exhibits a substantial decrease. The high degree of vegetation coverage of the study area before the 12 May 2008 Wenchuan Earthquake (92%) resulted in a strong positive correlation between the area of vegetation damage and the area of geohazards occurrence. In this study, we used fitted NDVI time series (MOD13Q1) which were able to determine the contribution of "non-geo-hazards" which were also responsible for a decrease in NDVI. In addition, we used the quantitative threshold method to extract the AVD caused by large- and medium- scale geohazards (LMG). Finally, we characterized the decreasing trend of geohazards in the study area of 10 counties/cities, paying special attention to five localities with strong fault activity. Our principal findings and conclusions are as follows:

(i) In the case of the 12 May 2008 Wenchuan Earthquake, previous research has examined the post-earthquake evolution of geohazards within localized areas or even within the entire earthquake-stricken area, based on studies of small watersheds within the most severely affected areas. The methodology used was to compare the yearly output volume of debris flows after heavy rainfall events for several years with the total debris volume of landslides induced by the 12 May 2008 Wenchuan Earthquake. Within the study area, there is a heterogeneous spatial distribution of geohazards in different tectonic environments, and in addition rapidly increasing land stability has resulted from the increasing coarseness of the residual debris material following the gradual loss of fine-grained landslide material. For these reasons, previous work [24] may have overestimated the duration of the process of evolution of geohazards. We attempted to avoid the limitations of generalizing from local estimates to the entire earthquake-stricken area by using a quantitative analysis of AVD evident in RSI. This approach is advantageous in terms of its large spatial scale and high degree of temporal resolution.

(ii) Our methodology was based on fitted MODIS-NDVI time series data, extracting the area of AVD caused by LMG, and using a quantitative NDVI threshold value which minimized the influence of non-earthquake factors (e.g., human activity and weather). This approach enabled us to characterize the decreasing trends of LMG in the study. However, due to the limitations

of the spatial resolution of the RSI, we were not able to distinguish between different types of geohazards or to detect any the evolutionary trends of small-scale geohazards.

(iii) Prior to the 12 May 2008 Wenchuan Earthquake (2001–2007), there was no significant evidence for a linear increase in the number of geohazards. Within this relatively stable geological environment, geohazards mainly occurred in alpine gorge areas or in areas with strong rock weathering. In addition to being affected topography and lithology, previous research has revealed that the occurrence of geohazards is positively correlated with rainfall [64,66]. The 12 May 2008 Wenchuan Earthquake was the principal factor triggering geohazards in the region. The seismogenic fault played the major role in determining the spatial distribution of geohazards: the closer to the epicenter, the more concentrated was the distribution of geohazards, and the more severe was the vegetation damage. Notably, the area of AVD in 2008 was 1–2 of magnitude greater than before the earthquake (during 2001–2007), which confirms that shallow earthquakes caused the most severe damage to vegetation. After the 12 May 2008 Wenchuan Earthquake (during 2008–2016), there was a well-developed spatiotemporal trend of decreasing geohazards in the area. After about 3 years following the 12 May 2008 Wenchuan Earthquake, the incidence of geohazards peaked slightly in 2011 before beginning to decrease. There were differences in the attenuation process of geohazards in different parts of the study region. The locality for Hongkou-Yinchanggou of Dujiangyan City and locality for Hongbai-Chaping of Shifang City were located in seismic intensity zone XI, where geohazard activity persisted until 2016, and the area of geohazards remained larger than that of the pre-earthquake level (2001–2007), compared with other localities in fault zones. In addition, we found that the higher the level of seismic intensity, the slower the rate of attenuation of geohazards. The trend of geohazards decrease in the southwestern part of the Longmenshan Fault Zone was slower than that in the northeastern part. The activity level of geohazards after the main shock in 2008 was not only correlated with rainfall, seismogenic faults and the seismic intensity of the 12 May 2008 Wenchuan Earthquake, but also with the number, magnitude and focal depth of earthquakes that occurred during 2009–2016. From the attenuation trend of AVD observed in this study, together with the results of previous research [18–22,24], we conclude that in the region with the highest level of seismic intensity (XI and X), geohazard activity will persist for 20–25 years; whereas in the region with lower seismic intensity (IX and VIII), geohazard activity will persist for ~15 years; and in the region of low seismic intensity (VII and VI), geohazards activity will be low.

**Supplementary Materials:** The MOD13Q1 images and the MODIS Reprojection Tool (MRT) software were downloaded from the website of https://lpdaac.usgs.gov/node/844 and https://lpdaac.usgs.gov/node/63, respectively. The Landsat-5 TM, Landsat-7 ETM+ and Landsat-8 OLI images were downloaded from the website of http://www.gscloud.cn/ and https://glovis.usgs.gov/ for free, the Gao-fen data were bought by business, parts of QuickBird data are from the Google Earth online Engine, the geohazards points data were collected from the Sichuan geological environment monitoring station (cutoff date 30 December, 2016). The earthquake data are from the website of https://earthquake.usgs.gov/earthquakes/map/.

**Author Contributions:** Z.N. performed the analyses and prepared the paper; Z.Y. provided the technical guidance and contributed to the design and discussion; W.L. provided the data of geohazards; Y.Z. helped the collection of Remote Sensing data and gave advice to this paper; Z.H. contributed to the error correction of typesetting and gave advice to this paper; All authors contributed to the experimental design, discussion and writing for the manuscript.

**Funding:** This work was supported by the Beijing post-doctoral fund (No. 2018-ZZ-096), the Capital Normal of University (No. 011185404207), the National Natural Science Foundation of China (NSFC) (No. 41521002), the Youth Fund of the Science and Technology Department of Sichuan Province of China (No. 2017JQ0031), and the Young and Middle-Aged Backbone Teacher Fund of Chengdu University of Technology (Sichuan Province, China) (No. 10912-2019KY51-01679).

**Acknowledgments:** The authors gratefully acknowledge Jan Bloemendal for his suggestions on this manuscript. We appreciate Weiguo Jiang and Kai Jia (Beijing Normal University) for their contributions to the mathematical formula derivation on this manuscript. We also thank all the anonymous reviewers for their valuable comments, which substantially improved our paper.

**Conflicts of Interest:** The authors declare no conflict of interest.

## Appendix A

The process of Equation (A6) as below:

The maximum of the curvature function of VDR is the threshold value *T*.

When the curvature (K) of the continuous X, it could express by Equation (A1).

$$K(x) = \frac{\left| f''(x) \right|}{\left( 1 + f'^2(x) \right)^{\frac{3}{2}}}, \tag{A1}$$

where, $f''(x) = \frac{df'(x)}{dx}$ and $f'(x) = \frac{df(x)}{dx} = \lim\limits_{\Delta x \to 0} \frac{\Delta y}{\Delta x}$ is the second derivative and first derivate of function $f(x)$. But the histogram of images is discrete, and the X is discontinuous. Then,

$$\Delta x = (x+1) - x = 1$$

where, the left image is the continuous function of independent variable X. The right image is the discrete function of histogram, f(x) is the function of histogram (Figure A1).

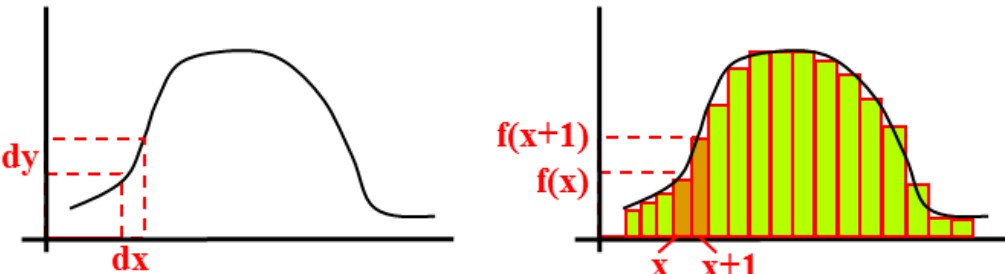

**Figure A1.** Continuous function (**left**) of variable X and discrete function of histogram (**right**).

The first derivate of the discrete function is the Equation (A2).

$$f'(x) = \frac{df(x)}{dx} = \lim\limits_{\Delta x \to 0} \frac{\Delta y}{\Delta x} = \frac{f(x+1) - f(x)}{1} = f(x+1) - f(x), \tag{A2}$$

The second derivate of the discrete function is the Equation (A3).

$$f''(x) = \frac{df'(x)}{dx} = \lim\limits_{\Delta x \to 0} \frac{\Delta f'(x)}{\Delta x} = \frac{f'(x+1) - f'(x)}{1} = f'(x+1) - f'(x), \tag{A3}$$

where, $f'(x) = f(x+1) - f(x)$, $f'(x+1) = f(x+2) - f(x+1)$;

Then, $f''(x) = f'(x+1) - f'(x) = (f(x+2) - f(x+1)) - ((f(x+1) - f(x))$;

So,

$$f''(x) = f(x+2) + f(x) - 2f(x+1). \tag{A4}$$

At last, put the first derivate of the discrete function of Equations (A2) and (A3) into the curvature function of Equation (A1), then the Equation (A1) could be expressed by Equation (A5).

$$K(x) = \frac{\left| f''(x) \right|}{\left( 1 + f'^2(x) \right)^{\frac{3}{2}}} = \frac{\left| f(x+2) + f(x) - 2f(x+1) \right|}{\left( 1 + (f(x+1) - f(x))^2 \right)^{\frac{3}{2}}}, \tag{A5}$$

In this paper, $x$ represents the $VDR$, $f(x)$ is the function of histogram. So in the text, Equations (A2) and (A4) expressed by Equation (A6).

$$Curvature(VDR) = \frac{|f''(VDR)|}{(1 + f'^2(VDR))^{\frac{3}{2}}} = \frac{|f(VDR + 2) + f(VDR) - 2f(VDR + 1)|}{(1 + (f(VDR + 1) - f(VDR))^2)^{\frac{3}{2}}}, \tag{A6}$$

where,

$$VDR_{i+1} = \frac{FVC_i - FVC_{i+1}}{FVC_i} \times 100\%, \tag{A7}$$

Here, $VDR_{i+1} > 0$ indicates that the $FVC$ value in year $i + 1$ is lower than $i$, and $i$ represents the year from 2001 to 2016.

The area of decrease $FVC$ are composed of the natural degeneration and the AVD caused by geohazards.

The pixel frequency distribution is described by the $VDR$ histogram, where the curvature of each point indicates a pixel change (Figure A2). An increased curvature indicates more severe change, therefore, the breakpoint between natural degeneration and the AVD caused by geohazards is the point where the curvature is maximal in the area with vegetation damage. In addition, because of disturbance from the image noise, there may be a pseudo breakpoint that is usually distributed on the two sides of the histogram. It can be excluded by the standard deviation of $VDR(\sigma)$. Most of the area with vegetation damage will fall within $k$ multiply $\sigma(3\sigma, k = 3)$. The $VDR$ corresponds to the maximum curvature is the required threshold $T$, and the maximum curvature in the histogram can be measured in the open interval of $(\overline{VDR}, \min\{\overline{VDR} + 3\sigma, VDR_{max}\}), k = 3)$ (the average of $VDR(\overline{VDR})$, the maximum of $VDR(VDR_{max})$). In here, the part of histogram for $VDR > 0$ is only considered. So the

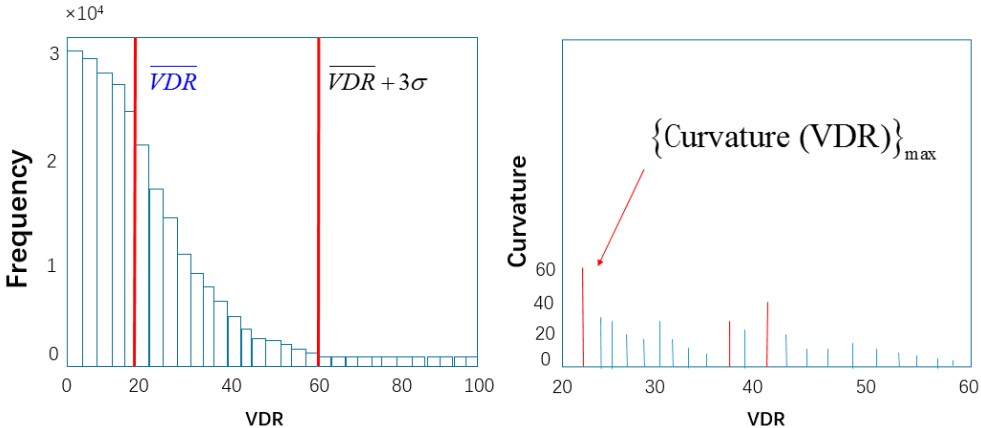

**Figure A2.** Frequency function of $VDR$ histogram (**left**) and Curvature function of $VDR$ histogram (**right**).

$T = (\{curvature(VDR)\}_{max}, VDR \in (\overline{VDR}, \min\{\overline{VDR} + 3\sigma, VDR_{max}\}), k = 3)$, the area with vegetation damage can be obtained by the threshold $T$.

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
