# Peer review of "Decreasing Trend of Geohazards Induced by the 2008 Wenchuan Earthquake Inferred from Time Series NDVI Data"

_remotesensing, doi:10.3390/rs11192192_

Round 1
Reviewer 1 Report
The article is interesting but contains some inaccuracies that should be corrected: Line 165. On the figure there are 5 localities (not 10)2.The reviewer is not sure if the presented results are correct just because they doesn't correspond with a pictures. Description seems to be for a different picture. The caption under figure 3 does not match the drawing. The reader has the impression that the interpretation of the results applies to other studies not shown in the drawings.
3. Please indicate the moment of earthquake on the fig. 4.
4. What is the difference between figures 4,5 and 6. (the captions are the same and the charts are different)
5. Total area of AVD should be listed for the same periods for example for one year or cumulative 2001-2002, 2001-2003, 2001-2004.
Specific comments
1.
In the list of literature there are many errors involving
the use of a lowercase letter at the beginning of
words that mean e.g. country, city, etc.
2. Sometimes Authors use different font e.g line118 "I/II"
3. line 141 capital letter
4. The Authors express large numbers using a dot and comma at the same time. In certain situations, this creates doubts in the interpretation of e.g. line 145 30,000 km2.
Author Response
Comments and Suggestions for Authors
Point 1: The article is interesting but contains some inaccuracies that should be corrected: Line 165. On the figure there are 5 localities (not 10)
Response 1: Thank you for this comment. “10 counties/cities” has been replaced by “10 counties/cities as the research area and 5 localities as the key studied areas” (Line 165-166).
Point 2: The reviewer is not sure if the presented results are correct just because they doesn't correspond with a pictures. Description seems to be for a different picture. The caption under figure 3 does not match the drawing. The reader has the impression that the interpretation of the results applies to other studies not shown in the drawings.
Response 2: Thank you for this comment. Figure 3 shows the process of verifying the results (Line 395-400). The extracted AVD are verified by 3 time-phases remote sensing images with relatively higher spatial resolution than MODIS before and after the occurrence of geohazards. The description of Figure 3 should be more detail.
“Maps of (a) in the left, middle and right show the same location, from the visual interpretation of Landsat-5, red color represents the areas covered by vegetation, the bright gray color represents geohazards such as landslides and collapses. The green polygons represent the extracted AVD by MODIS. The left maps in (a) and (d) represent the areas do not have geohazards, the middle maps in (a) and (d) represent the areas have occurred geohazards, the right maps in (a) and (d) are used to verify the correctness of the extraction results in 2006 and 2016, respectively. The left map of (c) represents the area of geohazards is relatively small than the middle map of (c), the right map of (c) is used to verify the correctness of the extraction results in 2011.”
I modified the related expressions in the title of Figure 3 on the original manuscript, and have submitted the revised manuscripts with labels of modifications.
Point 3: Please indicate the moment of earthquake on the fig. 4.
Response 3: Thank you for this comment. The expressions of “2008 Wenchuan earthquake” has been revised to “May 12, 2008 Wenchuan earthquake”. (The original expression is in Line 444, 454, and 463. Now is in Line 451, 461 and 470.)
Point 4: What is the difference between figures 4, 5 and 6. (The captions are the same and the charts are different)
Response 4: Thank you for this comment. Figures 4, 5 and 6 expressed 3 main characteristics which are delineated in detail in the original line 439-461, now is in line 446-468. The captions of the 3 figures are the same, but the charts are different and also have different meanings due to different localities of NDVI pixels.
Figure 4 shows the minimal change in NDVI before and after the May 12, 2008 Wenchuan earthquake.
Figure 5 shows the maximum value of NDVI decreased sharply in 2008, as a result of the May 12, 2008 Wenchuan earthquake , and dropped again in 2009 and 2010, then increased gradually in the following years.
Figure 6 shows the maximum NDVI value decreased slightly in 2009 and then increased gradually. This reflects the minor influence of the May 12, 2008 Wenchuan earthquake.
I added the expressions of “in different localities of NDVI pixels” in line 440 before the Figures 4, 5 and 6 on the original manuscript, and have submitted the revised manuscripts with labels of modifications.
Point 5: Total area of AVD should be listed for the same periods for example for one year or cumulative 2001-2002, 2001-2003, 2001-2004.
Response 5: Thank you for this comment. I modified the expression of “Total area of AVD” as “the total cumulative area of AVD” in Table 1 and the related expression on the original manuscript, and have submitted the revised manuscripts with labels of modifications.
Specific comments
Point 1: In the list of literature there are many errors involving the use of a lowercase letter at the beginning of words that mean e.g. country, city, etc.
Response 1: Thank you for this comment. I reset the reference management software and rechecked the letter of related words which need an uppercase in the list of literature.
Point 2: Sometimes Authors use different font e.g line118 "I/II"
Response 2: Thank you for this comment. I modified the font of "I/II" and the font of these similar expressions.
Point 3: line 141 capital letter
Response 3: Thank you for this comment. I modified the lowercase letter of “s” to uppercase letter of “S” in this initial letter.
Point 4: The Authors express large numbers using a dot and comma at the same time. In certain situations, this creates doubts in the interpretation of e.g. line 145 30,000 km2.
Response 4: Thank you for this comment. I carefully modified all the expression of large numbers using comma, and have submitted the revised manuscripts with labels of modifications.

Reviewer 2 Report
This manuscript investigates the spatial variation of AVD and its temporal evolution from NDVI values. The manuscript is well organized and well written, and I like the fundamental philosophy of the methodology. Thus I do not have too much concern, but my comments listed below will significantly improve the manuscript.
1. If I understand correctly, AVD is defined as the (spatial) curvature of VDR. It should be a good idea to define the boundary of AVD and non-affected areas, but I am afraid that the method misidentifies the central part of AVD because the curvature of AVD is not necessarily significant in the middle of AVD. I may be wrong because Figure 7 looks reasonable, but I would like the authors to add some sentences to disentangle my misunderstanding if any.
2. Figure 7 shows the temporal evolution of AVD by averaging images through the year (if I understand correctly). It is good, but AVD should be seasonally modulated because of the seasonal variation of precipitation. I would like to see how AVD changes monthly.
3. Line 50 (1984): Omori (1894)?
Reference:
Omori, F. (1894) On the Aftershocks of Earthquakes. Journal of the College of Science, Imperial University of Tokyo, 7, 111-120.
4. Although the term AVD is defined in Abstract, it should be defined in the first appearance in the main text (line 94).
5. Figure 2: It would be more helpful to readers if the authors unabbreviate MVC and FVC in the figure or the caption, although they are defined in the main text (lines 298 and 316).
Author Response
Response to Reviewer 2 Comments
Comments and Suggestions for Authors
Point 1: If I understand correctly, AVD is defined as the (spatial) curvature of VDR. It should be a good idea to define the boundary of AVD and non-affected areas, but I am afraid that the method misidentifies the central part of AVD because the curvature of AVD is not necessarily significant in the middle of AVD. I may be wrong because Figure 7 looks reasonable, but I would like the authors to add some sentences to disentangle my misunderstanding if any.
Response 1: Thank you for this comment. You understand correctly. I agree with your viewpoint. From a statistical point of view, there is indeed a situation that the small parts of AVD are misidentified by their curvatures are not significant in the middle of AVD. Therefore, this paper verifies the extraction results by comparing the visual interpretation results from high spatial resolution remote sensing images. This is also a limitation of this paper.
I added the expression of “By this paper’s extracted method, the small parts of AVD are misidentified by their curvatures are not significant in the middle of AVD due to some small-scale geohazards. Although this paper pays more attention to the law of the occurrence of large and medium-scale geohazards, these small geohazards are also worthy to be paid attention to, especially when the frequency of small-scale geohazards is high in local areas. In order to reduce these misidentifications, this paper verifies the extraction results by comparing the visual interpretation results from high spatial resolution remote sensing images.” for the limitations in Line 651.
Point 2: Figure 7 shows the temporal evolution of AVD by averaging images through the year (if I understand correctly). It is good, but AVD should be seasonally modulated because of the seasonal variation of precipitation. I would like to see how AVD changes monthly.
Response 2: Thank you for this comment. Changes of AVD are mainly consistent with the changes in seasonally precipitation. The previous literatures before the May 12, 2008 Wenchuan Earthquake confirmed the viewpoint that the high positive correlation between the occurrence of geohazards and the high amount of precipitation. After the May 12, 2008 Wenchuan earthquake, the precipitation and the aftershocks have combined to act on the occurrence of geohazards (this paper’s viewpoint).
Theoretically, the monthly changes of AVD could be calculated by MODIS-NDVI. However, MODIS-NDVI has 2 data per day. In fact, not every day has the reliable data due to clouds contamination. The 8-day data products of MODIS-NDVI are produced which still cannot satisfy the practical application. Then, the 16-day or the monthly data products of MODIS-NDVI are produced. In this paper’s approach, in order to improve the accuracy, the MODIS-NDVI of the vegetation growing season (May to September) through the year are selected to calculate the AVD by making up for the lack of data as much as possible. If only the monthly data products of MODIS-NDVI are used to calculate the AVD, the accuracy of the results is doubtful due to the outliers or missing data.
Point 3: Line 50 (1984): Omori (1894)?
Reference: Omori, F. (1894) On the Aftershocks of Earthquakes. Journal of the College of Science, Imperial University of Tokyo, 7, 111-120.
Response 3: Thank you for this comment. I modified the expression of “Omori (1894)” in a correct reference.
Point 4: Although the term AVD is defined in Abstract, it should be defined in the first appearance in the main text (line 94).
Response 4: Thank you for this comment. I modified the expression of AVD in the first appearance in the main text in Line 94.
Point 5: Figure 2: It would be more helpful to readers if the authors unabbreviate MVC and FVC in the figure or the caption, although they are defined in the main text (lines 298 and 316).
Response 5: Thank you for this comment. I modified the expression of “MVC” and “FVC” using the unabbreviated forms in Figure 2.
